# Diagnosis of *Trypanosoma cruzi* infection in Mexican populations: current conventional serology lacks adequate sensitivity and specificity

Janine M Ramsey[1], Keynes de la Cruz-Felix[1], Ezequiel Tun-Ku[1], Alejandro G Schijman[2], Sleidher Gutiérrez[1], Margarita Virgen-Cuevas[3], Monica Reyes-Romero[3], Kenia Escobedo-López[3], Gilberto Sánchez-González[4], Angélica Pech-May[1]/+

[1]Centro Regional de Investigación en Salud Pública, Instituto Nacional de Salud Pública, Tapachula, Chiapas, México
[2]Instituto de Investigaciones en Ingeniería Genética y Biología Molecular Dr Héctor Torres, Consejo Nacional de Investigaciones Científicas y Técnicas, Argentina
[3]Instituto Nacional de Ciencias Médicas y Nutrición Salvador Zubirán, México City, México
[4]Centro de Investigaciones en Enfermedades Infecciosas, Instituto Nacional de Salud Pública, Cuernavaca, Morelos, México

**BACKGROUND** The performance of serological tests for *Trypanosoma cruzi* diagnosis in Mexico has not included discordant control sera nor has it evaluated the role of immune response specificities, patient infection history or clinical status.

**OBJECTIVES** The performance of commercial serological and molecular diagnostic tests and diagnostic algorithms was analysed in Mixtecan and Zapotecan ethnic populations having recent and long-term infection history.

**METHODS** An amplified global gold standard for *T. cruzi* infection included serological ($\geq$ 2 conventional tests positive) and molecular (sequence identity of any of five genes using end point polymerase chain reaction (epPCR) or any positive using quantitative polymerase chain reaction (qPCR) diagnostic test results.

**FINDINGS** Only 81% of previously diagnosed untreated infections were reconfirmed using serology, while an additional 14% only using PCR. Serological diagnosis sensitivity ($\geq$ 2 tests positive) in the primary diagnosis cohort was 8%, while specificity was 16%. Diagnosis sensitivity was similar using epPCR and qPCR only in primary diagnoses and all identified using the satellite (SAT) gene. The 18S ribosomal DNA identified *T. cruzi* and *T. dionisii* co-infections from Pacific coast sites.

**MAIN CONCLUSIONS** The current study provides evidence for inadequate diagnostic performance of conventional serological tests and the need to develop appropriate antigenic tools and use molecular testing of seronegatives to ascertain absence of infection.

Key words: Chagas disease - *Trypanosoma cruzi* - serological diagnosis - performance - molecular diagnosis

Zoonotic *Trypanosoma cruzi* and Chagas disease in humans (CD) are widely distributed in the American continent as in Mexico, causing important clinical and economic burden.[1,2,3,4] Estimates of CD prevalence in Mexico suggests that between 2 and 4 million individuals may be *T. cruzi* infected and that incidence has doubled from 46,600 cases in 1990[5] to 86,500 cases in 2018.[6,7] These data, however, are considered underestimates since there has been systemic deficiencies in acute and chronic case detection, treatment and transmission prevention, at least prior to 2020.[8]

Seroprevalence analyses covering most regions in Mexico suggest gradients among urban, suburban, and rural areas such as 1.0%, 2.4% to 3.2%, respectively, from Nuevo Leon,[9] while 6.6% from rural Queretaro,[10] 17.8% from rural Jalisco[11] and 4.8% from rural Campeche.[12] Migrant Mexican blood donors double-tested in the United States (not filtered by single test screening) are 2.2% to 4.7% seropositive,[13] although blood samples have low antibody reactivity and clinical sensitivity.[14]

Analyses of 64,969 blood donations from 18 Mexican states reported 1.5% seropositivity overall in blood donations (subjected to screening filters),[15] although other studies report a range 0.01% to 3.1% seropositive screened donations[16] and independent blood bank studies report from 0.5% to 2.8% seroprevalence.[17,18] Even assuming no impact due to screen test false seronegativity, a blood bank survey conducted in 2008 (560 blood banks) estimated approximately 2000 inhabitants each year are at risk of contracting *T. cruzi* infection due to the lack of universal donation blood screening,[19] while a subsequent study (503 blood banks) estimated that only 74% of banks conducted quality control, suggesting continued deficient screening compliance.[20] Improper

Financial support: Conacyt SEP (grant #166828), FONSEC (grant #161405, #261006, the project grants 50-63450, 50-6199), Fondo Nacional de Promoción para la Salud de la Secretaria de Salud (JMR), Ministery of Science Technology and Innovation, Argentina, PICT 2020-0862 (AGS).
+ Corresponding author: apechmay@gmail.com
https://orcid.org/0000-0003-4942-7788

undisclosed performance analyses for screening procedures may cause the bias observed in the discrepancies evidenced between seroprevalence reported by blood banks vs. open populations. A 4% seroprevalence is reported from rural populations in Puebla,[21] although blood donation seroprevalence ranges from 0.14% in an urban centre,[16] 1.24% in rural, but 7.7% in suburban banks.[22,23] In Veracruz, seroprevalence in blood donations ranges from 0.5%[18] to 0.94%,[16] while there is 16.8% seroprevalence evidenced from 19 rural communities.[24] A similar example from Chiapas reports 0.31% seroprevalence from urban blood donations,[16] while 22% seroprevalence reported from rural populations[25] and 5.0% in pregnant women.[26]

Although these discrepancies may derive from multiple study design and testing biases and/or different procedures used for diagnosis,[27,28,29] including the double urban to rural blood donor ratio, other poorly analysed factors such as *T. cruzi* population shifts over the course of natural infections and differences of immunodominant phenotypes from short and long-term infections may affect the quality, quantity or affinity of antibody responses.[30,31] Phenotypic and antigenic differences have been evidenced between *T. cruzi* strains from South and North America and there are clear differences in immune responses among human infections from different regions within the continent.[32,33] Additionally, multiple clinical case reports in Mexico have reported false negative serology from clinical settings in patients with demonstrated *T. cruzi* infection using histology,[34] mini-exon (ME) end point polymerase chain reaction (epPCR) and haemoculture,[35] PCR using brain tissue biopsy[36] and using satellite (SAT) and 18S ribosomal DNA (18SrDNA) PCR.[37] Given high *T. cruzi* diversity evidenced in Mexican vectors and zoonotic reservoirs,[38,39,40] it is expected that even within the country there may be a broad diversity of immune responses, despite the dominant discrete typing unit I (DTUI) lineage and secondary prevalence of DTUVI.[41] Non-autochthonous recombinant, synthetic antigens or proteins/peptides may not reflect regionally-specific parasite strains, diversity or shifting specificities. Epitope specificities and low-level *T. cruzi* specific antibodies may not be detectable using current conventional or point of care rapid tests, thereby reducing sensitivity and specificity to detect human immune responses in Mexico.[27,31,42,43] Low sensitivity (62.5%) of the Stat-Pak rapid test as compared to conventional assays and in comparison with South American populations of seropositive pregnant women is another clear example of regional antigen/immune response variation.[44]

Precise diagnostic capacity of *T. cruzi* infection is essential to characterise the diversity and broad distribution of CD burden and inform public policy in Mexico, whether by serological or parasitological methods. In order to evaluate the performance of current diagnostic tools and algorithms for population cohorts having different infection history, the aim of the present study has been to analyse the sensitivity and specificity of currently available *T. cruzi* serological and molecular diagnostic methods using a global gold standard. Herein we report results from previously studied population cohorts from three regions in Oaxaca. Two groups from each site were selected: the first was composed of individuals who had previous *T. cruzi* diagnosis but no etiological treatment (secondary diagnosis group - SDG) and the second was a same-community control of exposed subjects diagnosed for the first time (primary diagnosis group - PDG). Samples from all subjects were analysed using multiple serological tests currently approved in Mexico (conventional and rapid), recombinant *T. cruzi*-specific proteins, and molecular analyses using epPCR (and sequencing of five nuclear and mitochondrial gene fragments) and TaqMan quantitative polymerase chain reaction (qPCR) for two genes.

## MATERIALS AND METHODS

*Ethics statement* - Studies were conducted in accordance with the Declaration of Helsinki, and the protocol was approved by the Instituto Nacional de Salud Pública de México Ethics Committee (CI1237 to JMR). All adult subjects or parents-guardians of minors gave their signed informed consent and minors signed assent for participation prior to inclusion in the study and all provided approval for secondary use of remaining blood sample aliquots. No specimen is connected to any personal information. Participants were also invited and signed informed consent to be evaluated by a project physician through a clinical history, general physical examination, and 12-lead electrocardiogram. All diagnostic and clinical results were given to participants individually (on paper and electronically if requested) and overall results and strategies to access healthcare and treatment through the public health service (PHS) discussed in collective meetings. All diagnostic and clinical results were submitted to the corresponding Sanitary Jurisdiction Heads (JSI, JSII, JSIV in Oaxaca). There are no conflicts of interest to report.

*Study sites and populations* - Study populations were invited to participate during county and community-based CD longitudinal engagement, surveillance and prevention projects conducted between 2003 and 2015 in three distinct regions in Oaxaca state: (1) Santos Reyes Nopala (SRN) county (0970837 latitude/160621 longitude, 45 communities), (2) Salina Cruz (SC, 0951145 lat/161057 long), and (3) Santa Cruz Papalutla (SCP, 0963502 lat/165720 long) [Supplementary data (Fig. 1)]. Demographic and exposure variables for each of the three populations are summarised in Supplementary data (Table I).

In each original engagement project, any individual recognising direct contact with bugs based on self-evaluation (having seen bugs in houses, having had dermal chinchomas or having altered cardiological or gastrointestinal symptoms) requested and was tested for *T. cruzi* infection. Following signed informed consent and assent (minors), blood samples were analysed using two-three *T. cruzi* serological tests by each of three laboratories (the Oaxaca State Public Health Laboratory, the National Diagnostics and Reference laboratory, and the Instituto Nacional de Salud Pública – INSP - Chagas laboratory). Sustainability of community-based surveillance and prevention was evaluated periodically and in 2015 none of the 45 inhabitants originally diagnosed had been treated: SRN (N = 12),

SCP (N = 19), or SC (N = 14). All inhabitants originally diagnosed with *T. cruzi* (between 2004 and 2008) were assigned to the SDG and any family member or neighbour having exposure risk or seeking diagnosis between 2016-2017 was invited to participate and assigned to the PDG. Following informed consent and approval to share diagnostic and clinical results with public health services for subsequent treatment, all participants responded to an exposure risk survey and a blood sample (two vials of 5 mL) was drawn for standard serology (four tests) and one rapid serological test (one vial in serum separation tube - SST) and the second vial in guanidine - EDTA buffer (GEB) for molecular parasite detection.[38]

*Serological diagnosis using conventional and rapid test* - Serum samples were tested for *T. cruzi* antibody using four conventional commercial enzyme-linked immunosorbent assay (ELISA) included in Mexican and international reference strategies. All assays were conducted in the Infectious Disease Laboratory at the Instituto Nacional de Nutrición y Ciencias Médicas Salvador Zubirán (INNCMSZ) per manufacturer instructions: (S1) Bio-Rad Chagascreen Plus v4 (recombinant antigen), (S2) Test ELISA Chagas III, Bios, Chile, (S3) Accutrack Chagas Microelisa Test System, Laboratório Lemos S.R.L., (S4) Accutrack Chagas recombinant microELISA Test, Laboratorio Lemos S.R.L.[45] All serum samples were evaluated in duplicate and ELISA cut-off values were calculated as the mean optical density (OD) of replicate true negative serum samples plus three standard deviations (SD) of that mean (+ 3 SD). Aliquots of all serum samples were additionally analysed using the Chagas Ab Rapid serological test (Standard Diagnostics Bioline) according to the manufacturer's instructions.[46]

*Trypanosoma cruzi-specific antibody using recombinant proteins* - All sera were screened for antibodies reactive to a panel of 10 recombinant *T. cruzi* proteins (Kn122, Kn107, Kn80, Kn117, FF10, LE02, G10, Fab4, ATPase, calmodulin) in a Luminex-based format at the Centre for Tropical and Emerging Global Diseases, University of Georgia, as previously described.[28] All samples were run in replicate in four batches along with *T. cruzi* Y strain lysate (amastigote/trypomastigote) and green fluorescent protein (GFP)(negative control protein) to independently validate sample quality and absence of immune suppression to *T. cruzi* antigens. Results from any sample having GFP values above the negative mean + 3SD are not reported due to non-specific activity.

*Parasites* - The CARI06 (*T. cruzi* lineage I; TcI) and CL-Brener (*T. cruzi* lineage VI; TcVI) strains were grown in liver infusion tryptose (LIT) and the genomic DNA of seven log dilutions (5 x $10^3$ to 5 x $10^{-3}$ parasites/mL) of TcI or TcVI spiked negative blood for PCR titration series. *T. cruzi* titration series and negative blood controls were used to determine each gene fragment's sensitivity using epPCR and qPCR. In addition, Silvio X10 (TcI) and CL-Brener (TcVI) prepared by Instituto de Investigaciones en Ingeniería Genética y Biología Molecular (INGEBI) as previously reported for standard curves for all quantitative TaqMan PCR were run in parallel with CARI06 (TcI) spiked titration series.[47]

*Molecular detection of T. cruzi* - Genomic DNA from blood preserved in GEB was extracted using a phenol-chloroform protocol and paired extraction aliquots used for epPCR and qPCR to reduce methodological bias.[48] All samples were analysed for *T. cruzi* DNA by epPCR using five gene fragments: a 195 bp fragment of SAT DNA using primers cruzi1/cruzi2,[49] the S34/S67 fragment (120 bp) for the conserved repeat region of kinetoplast DNA (kDNA),[50] a fragment of the small subunit 18S ribosomal DNA gene (18S rDNA) amplified using SSU561F/SSU561R (560 bp), although all products between 550-750 bp were sequenced,[51] the spliced leader mini-exon gene (ME; 300-350 bp representing lineage I and II),[52] and the 24Sα ribosomal DNA (24S rDNA) gene using the D71/ D72 primers which, similar to the ME, produce alternative size bands depending on the lineage (110-130 bp).[53] DNA amplification protocols for epPCR were as previously described[47] and *T. cruzi* DNA from CARI06 and CL-Brener parasite strains and negative amplification controls were processed with test sample batches using epPCR; all samples not amplifying parasite controls were analysed using the cytochrome b (cyt *b*) gene for integrity.[50] Triplicate aliquots of spiked serial dilutions (5 x $10^3$ - 5 x $10^{-3}$ para/mL) of CARI06 and CL-Brener were also extracted and all five gene fragments amplified using epPCR. Relevant amplicon bands were sequenced from test and control samples to ascertain specificity and sensitivity limits of electrophoretic, purification and sequencing procedures. All amplicons of previously reported (expected) band size and range were purified using QIAquick Gel Extraction kits (QIAGEN, Valencia, CA) and capillary sequencing carried out on an Applied Biosystems 3730XLs (Macrogen, Korea). Internal plate controls for sequencing quality were included to monitor cross-contamination of amplicons among specimens and positive controls.

The TaqMan qPCR was conducted using cruzi1/cruzi2/cruzi3 primers for SAT DNA (SAT qPCR) and 32F/148R/71P primers for kinetoplast kDNA (kDNA qPCR) at INGEBI.[47,49] Paired aliquots of DNA extracted from all samples at CRISP/INSP and used for epPCR were transported in $H_2O$ to INGEBI for TaqMan qPCR processing. Aliquots of serial dilutions of spiked control blood samples of TcI (CARI06, Silvio X10) and TcVI (CL-Brener), the dominant DTUs expected, were used to compare dynamic ranges for the two primer sets and standard curve parameters were estimated for all primers and DTUs.

*Data analyses* - All samples were scored for *T. cruzi* infection based on results from (1) individual conventional serological tests: positive if mean OD ≥ OD negative+3SD, (2) rapid serological test: positive or negative, (3) reactivity to individual recombinant proteins and Y lysate: mean OD ≥ OD negative+3SD,[54] (4) *T. cruzi* sequence identity to GenBank registries of consensus sequences from expected size epPCR amplicons of the five gene fragments: SAT, kDNA, ME, 18S, 24S, and (5) TaqMan qPCR using SAT or kDNA: threshold counts (Ct) above positive control, positives were scored as either quantifiable or non-quantifiable.

Positive *T. cruzi* serological diagnosis was scored if ≥ 2 conventional serological tests were positive. *T. cruzi* infection (parasitaemia) using epPCR was scored positive if amplicons from ≥ one of five gene fragments had *T. cruzi* sequence identity and also using qPCR (SAT or kDNA), scored positive if the Ct value was ≥ to 0.01 parasite equivalents/mL, with quantitative values (Q), or below this threshold but above negative control as non-quantitative (NQ). The combined global infection gold standard (global diagnosis) scores any sample positive if there is a positive result by any one of the three diagnosis criteria: conventional serology (≥ 2 tests positive) and/or epPCR and/or qPCR.

The rapid serological test was not used for overall serological or global diagnosis, although its correlation and concordance with that of all individual conventional tests, as well as with serological diagnosis, was analysed using Pearson's correlation coefficient. Correlations (correlation matrix) of the different diagnostic test results were computed using Pearson's correlation coefficient to identify tendencies of results from the PDG and SDG separately. Serologic diagnosis, epPCR diagnosis, qPCR diagnosis and global diagnosis were also correlated separately for the PDG and the SDG and results of individual epPCR using SAT, kDNA or ME and qPCR with either SAT or kDNA were each correlated with global diagnosis. Sensitivity (true positive/true positive + false negatives) and specificity (true negative/true negative + false positives) of serological diagnoses, epPCR, and qPCR were calculated as the ratio of positive or negative results, using global diagnosis as reference for both the PDG and the SDG. The sensitivity and specificity of serological diagnosis to reconfirm the original diagnoses (SDG) was also calculated. The concordance between serological, epPCR, qPCR and global diagnoses was analysed using Cohen's kappa coefficient to control for randomness of results. Recombinant reactivity differences between the PDG and the SDG were analysed using a paired t test.

Gene amplification specificity for each gene fragment in epPCR (expected size amplicons) is calculated as the ratio of sequences with identity to *T. cruzi* to all size-relevant amplicons sequenced for the fragment. Standard curves for serial dilutions of TcI and TcVI were obtained by linear curve adjustment of the data. The curve-adjusting parameters (slope and intercept) were used to generate 30 random numbers distributed along the curves considering a normal distribution with a 10% standard deviation. This set of silico-generated data was used to calculate the mean difference and statistical significance using a paired t-test, considering a null-hypothesis of mean difference equal to zero. Adjusted parasite load was calculated for all qPCR quantitative results based on a factor calculated both for SAT and for kDNA from the spiked control standard curves for Silvio and CARI06 for SAT, and for CARI06 and CL-Brener for kDNA [Supplementary data (Fig. 2)]. Median parasite loads for PDG and SDG were compared separately for SAT and kDNA excluding high outliers for SAT (1,488 and 2 parasites/mL) and kDNA (5, 8, 10 parasites/mL).

Consensus sequences of the five gene fragments were generated independently using MEGA v.10.[55] Sequence identity for the five fragments was generated using the GenBank platform (https://blast.ncbi.nlm.nih.gov/), using priors with query cover values above 70% and percentage identity above 80%; the first 100 sequences were considered. Haplotypes and haplotype diversity were generated using the DnaSP v.5.10 software for each gene.[56] Representative *T. cruzi* haplotypes were deposited in GenBank: accession numbers for the SAT fragment: OR288103-OR288138; accession numbers for the kDNA fragment: OR288139-OR288142; accession numbers for the 18S fragment: OR286389-OR286392. Unique *T. cruzi* SAT haplotypes were used to analyse relationships and an independent median-joining haplotype network was generated using Network v.4.6.[57] Representative *T. dionisii* 18S haplotypes were also deposited in GenBank: accession numbers OR286393-OR286397.

## RESULTS

*Study populations* - Demographic variables for participants assigned either to the PDG or the SDG from each study site were not significantly different, with the exception of the SDG in the predominately rural SRN county (high male emigration) [Supplementary data (Table I)]: lower women/men ratio, average age and number of years of schooling. Despite differing elevations and surrounding topography, degree of urbanisation, total population, and predominant vector species, the three sites are located between 220 and 300 km in linear distance from each other [Supplementary data (Fig. 1)]. Highest positive predictive value for bug identification was from SC (*Triatoma phyllosoma* is the largest species), followed by SRN (*Triatoma mazzottii* similar size to *T. phyllosoma*, but the secondary *Triatoma dimidiata* Hg2 is smaller) and lowest in SCP, where one of the smallest of Mexican species is found (*Triatoma barberi*). The dermal inflammation caused by the bug bite ("chinchoma") and the memory of bugs in houses or being bitten were consistent with bug size.

*Parasite detection limits for five T. cruzi gene fragments using epPCR* - The parasite detection threshold using epPCR was lowest using the SAT ($5 \times 10^{-3}$ parasites/mL; para/mL) and kDNA ($5 \times 10^{-2}$ para/mL), while 5 para/mL for 18S, $5 \times 10^2$ para/mL for ME and $5 \times 10^3$ para/mL for 24S. Proportion of infections identified using each gene singly or in combination with another gene are summarised in Supplementary data (Table II). Replicate amplifications were run for control on 30% of samples for kDNA, 31% for the ME, and 55% for the 18S, resulting in complete concordance for amplification and identity results following sequencing. A total of 57 samples amplified for one or more of the five genes (58% one gene, 40% two genes), 95% of these using SAT (57% alone, 43% with at least a second gene), but only 26% using kDNA. The kDNA only identified one infection not amplifying with any other gene. Two infections not amplifying with either SAT or kDNA were identified only using the ME (lineage I).

Fragment amplification specificity, the proportion of expected-size amplicons having *T. cruzi* sequence identity, was different among sites and for each fragment, even though overall, there was high amplification rates using the primers for kDNA (99%), while lower for SAT, ME, and 24S (65%-70%) [Supplementary data (Table III)]. Amplicon specificity using SAT (88.5%) was significantly higher in the urban inland SCP than in SRN (P = 0.041) or SC (P = 0.021). Amplicon specificity for *T. cruzi* kDNA was significantly lower than SAT overall (17.2%), although in contrast to SAT, significantly higher in SCP (32.4%) than in SRN (7.4%, P = 0.005) or SC (4.3%, P = 0.001). The specificity of ME lineage I (ME LI) amplicons (350 bp) in SCP was significantly lower to that in SRN (26.7%, P = 0.02) or SC (25.0%, P = 0.02) and none of ME lineage II (ME LII, 300 bp, N = 33) or 24S amplicons (N = 59) from any site had sequence identity for *T. cruzi*.

*Parasite detection limits using SAT and kDNA TaqMan qPCR* - The sensitivity (parasite load) of the TaqMan qPCR using SAT and kDNA is 0.01 - 0.02 para equiv/mL for TcI (INGEBI).[47] Both SAT and kDNA were analysed using qPCR in 74% of samples, while sensitivity of single SAT analysis was 87% and the negative predictive value for SAT qPCR was 94%.

Parameters of the qPCR standard curves were evaluated using both INGEBI-standard spiked TcI and TcVI control samples and the in-house TcI-spiked 10-fold dilution series (CARI06) (Fig. 1). There were significant differences between SAT dynamic ranges of the Mexican TcI (CARI06) vs. Silvio X10 TcI (P = 3.8 x E-08) and vs. CL-Brener TcVI (P = 5.0 x E-03). TcI-CARI06 kDNA Ct values were significantly lower than TcVI CL-Brener (P = 5.7 x E-19) and therefore positive Ct values below external control test laboratory limits (INGEBI) were scored positive as not-quantifiable (NQ).

*Conventional and rapid test serology* - Overall, 65.9% of the SDG were reconfirmed using conventional serology, while an additional 29.6% were confirmed only using PCR (total 95.5% infections reconfirmed; Table I). Overall, 4.5% of the PDG were *T. cruzi* positive using conventional serology, although 86.4% were infected using either PCR method. Conventional serological diagnosis (≥ 2 tests positive) of the PDG had low correlation with global infection diagnosis (0.107) and that of the SDG was random (-0.433) (Table IIA). Sensitivity of conventional serology to reconfirm previous diagnosis was 48% [95% confidence interval (CI): 0.31, 0.65], while sensitivity for serological diagnosis of the PDG was 8% (95% CI: 0.0, 0.17) (Table IIB). Specificity of serological diagnosis indicated low confidence in seronegative results with 16% for the PDG (95% CI: 0.04, 0.28) and 13% for the SDG (95% CI: 0.01, 0.25), which was additionally evidenced by low concordance of serological and global diagnoses: 20.4% (K = -0.30, -1.16, 0.55) for the PDG and 31.8% (K = 0.94, 0.31, -1.57) for the SDG (Table IIC). Concordance between original diagnosis and serological reconfirmation (of SDG) was 36.3% (K = -0.008, 95% CI: -0.63, 0.61).

The correlation matrix for all individual conventional serological tests was consistently linear, indicating a high probability of similar results between tests and the rapid test (80.7% and 90.9%, respectively) for both the PDG and the SDG [Supplementary data (Table IV)]. The correlation of global diagnosis with individual conventional serological or rapid tests and serological diagnosis was similarly low for the PDG (10.7% and 8.7%, respectively) and random for the SDG (-0.433 and -0.352, respectively).

*Serological diagnosis sensitivity differences among sites* - Serological sensitivity was heterogeneous among sites; no infections were reconfirmed (SDG) using conventional serology from SC, while 81% from SRN and 89% from SCP were reconfirmed. However, all seronegative infections from SRN (18.7%) and SC (N = 17) were reconfirmed using PCR (Table I). The proportion of infections from both the SDG and the PDG identified by both serological and molecular techniques was also significantly different among sites (t = -5.37, df = 115, P = 2.00x10⁻⁷).

Two discordant serology cases were both confirmed only using PCR. One previously diagnosed case with original discordant serology (SC #11, 2006) was seronegative again in two new serial samples, but *T. cruzi* positive using epPCR (SAT, ME, and 18S) and qPCR (SAT

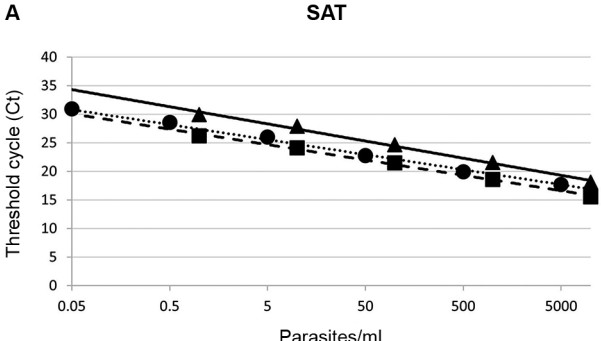

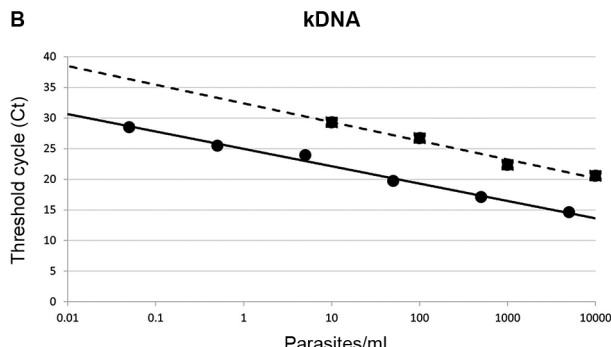

Fig. 1: standard curves for serial dilutions of *Trypanosoma cruzi* lineage I (Silvio X10 and CARI06) and *T. cruzi* lineage VI (CL Brener) in TaqMan polymerase chain reaction (PCR) assay using satellite (SAT) DNA (A) and kinetoplast DNA (B) primer sets. Standard curve parameters (slope/ intercept/ r2) for SAT DNA were: Silvio X10/INGEBI (triangle, -1.304/30.43/0.0029), CARI06/INSP (circle, -1.141/27.41/0.9961), CL Brener/INGEBI (square, -1.168/26.58/0.9956), and for kinetoplast DNA (kDNA): CARI06 /INSP (circle, -1.231/24.96/0.9941), CL Brener/INGEBI (square, -1.325/32.37/0.9759).

## TABLE I

Serological and molecular diagnosis for *Trypanosoma cruzi* infection in patient samples of primary diagnosis group (PDG) and secondary diagnosis group (SDG) from three counties in Oaxaca

| Site | Previous *T. cruzi* testing | Number samples (N patient) | Serology | | | | | Molecular | | | Global infection *T. cruzi* (N) |
|---|---|---|---|---|---|---|---|---|---|---|---|
| | | | *T. cruzi* positive - 4 conventional tests (N) | *T. cruzi* positive RapT (N) | Mean num. reactive RP (N) | Mean ± SD OD Lysate (N) | Lysate/ GFP | epPCR (N) | qPCR (N) | *T. cruzi* serology & molecular (N) | |
| Salina Cruz (SC) | PDG | 13 (13) | 0 (13) | 0 (13) | 9.3 (11) | 2494 ± 1897 | 1.7 | 76.9% (13) | 100%[b2] (13) | 0 (13) | 100% (13) |
| | SDG[c] | 2 (1) | 0 (2) | 0 (2) | 10 (2) | 4076 ± 388 | 2.0 | 100% (2) | 50.0% (2) | 0 (2) | 100% (2) |
| | SDG[d] | 8 (3) | 0 (8) | 0 (8) | 9.9 (8) | 6445 ± 5968 | 5.3 | 100%[a5] (8) | 50.0%[b1] (8) | 0 (8) | 100% (8) |
| Santos Reyes Nopala (SRN) | PDG | 12 (12) | 16.7% (12) | 16.7% (12) | 9.3 (11) | 6233 ± 7518 | 9.3 | 83.3%[a4] (12) | 100.0%[b2] (12) | 16.7% (12) | 100% (12) |
| | SDG | 16 (15) | 81.3% (16) | 75.0% (16) | 9.8 (15) | 16243 ± 6406 | 9.8 | 75.0%[a6] (16) | 68.8% (16) | 68.8% (16) | 100% (16) |
| Santa Cruz Papalutla (SCP) | PDG | 19 (19) | 0 (19) | 0 (19) | 9.8 (19) | 4028 ± 4077 | 3.9 | 63.2% (19) | 36.8%[b5] (19) | 0 (13) | 68.4% (19) |
| | SDG | 18 (18) | 88.9% (18) | 77.8% (18) | 9.9 (18) | 10304 ± 7114 | 6.6 | 16.7% (18) | 16.7%[b2] (18) | 18.8% (16) | 88.9% (18) |
| Oaxaca | PDG | 44 (44) | 4.5% (44) | 4.5% (44) | 9.5 (43) | 4167 ± 5028 | 4.7 | 72.7% (44) | 72.7%[b5] (44) | 5.3% (38) | 86.4% (44) |
| | SDG | 44 (37) | 65.9% (44) | 59.1% (44) | 9.9 (43) | 11724 ± 7614 | 10.7 | 56.8% (44) | 43.2%[b5] (44) | 38.1% (42) | 95.5% (44) |

Global infection is the sum of serology, end point polymerase chain reaction (epPCR) and quantitative polymerase chain reaction (qPCR) results; RP: recombinant proteins (N = 10). a: # samples also with *T. dionisii* by 18S epPCR; b: # NQ *T. cruzi*, all positive also by epPCR, serology or qPCR; c: only 1 test positive; d: two tests positive in three laboratory.

and kDNA). The same individual was also coinfected with *T. dionisii* based on identity of 18S sequences. In addition to this patient, another individual from the PDG was seropositive in only one of four conventional serological tests, but *T. cruzi* infected using SAT qPCR.

*Trypanosoma cruzi*-specific antibody to recombinant proteins - Sera from 79 patients (N = 86, two samples excluded due to high reactivity to the GFP control) from all sites and both diagnostic groups reacted each to between nine and 10 recombinant proteins (Table I). The mean Y lysate reactivity was significantly higher in SDG vs. PDG in all three sites (P < 0.05) and that of the SDG was significantly different among the three sites (P < 0.002). Reactivity to all recombinant proteins, previous and current serological diagnosis category, diagnostic group, current global diagnosis (serology+epPCR+qPCR), Y lysate/GFP ratio, and identification of single vs multiple-samples for patients from the three sites are summarised in a heatmap (Fig. 2). Despite heterogeneous responses, highest reactivity (> 95% CI) was observed unexpectedly in seropositive samples which were also PCR positive, as seen in the right half of the heatmap along with the positive test control (PTC). Samples from coinfected *T. cruzi-T. dionisii* had both high (#55, 56, 59, 64) and low (#1, 2, 70) recombinant protein reactivities. There was a range of serum reactivity profiles, although highest responses were to proteins Kn122, FF10, Kn107, Fab4, LE2, Kn80 and the Y strain lysate (Fig. 3A). Reactivity was significantly higher in the SDG as compared to the PDG for Kn107 (P = 0.0001), Fab4 (P = 0.006), LE2 (P = 0.0002), calmodulin (P = 0.04), G10 (P = 0.01), and the Y lysate (P = 0.0001) (Fig. 3B).

*Trypanosoma cruzi* infection using PCR - Infection prevalence overall was similar between the two PCR methods for both PDG and SDG (Table I). Despite poor correlation between epPCR and qPCR overall, 31.3% for the PDG and 38.9% for the SDG, the correlation of global diagnosis with epPCR and qPCR was the same for PDG (65%), although higher using epPCR vs qPCR in the SDG (100% and 39%, respectively) (Table IIA). The sensitivity of epPCR and qPCR was 84% (95% CI: 0.72, 0.96) for the PDG, while 100% using epPCR and 60% (95% CI: 0.44, 0.76) using qPCR for the SDG (Table IIB). The specificity using both epPCR and qPCR was 50% for the PDG, while 100% using epPCR and 60% using qPCR in the SDG. Overall, most *T. cruzi* infections from the PDG (95%), while only 31% from the SDG were diagnosed only using PCR (Table III). All the PDG and 88% of the SDG infections using epPCR were identified using the SAT gene; two or more genes amplified from 53% of the PDG, but only 28% of the SDG infections.

Concordance of serological diagnosis using qPCR was greater than using epPCR, although it was similar for the PDG (34.0% and 29.5%, respectively) and the SDG (54.5% and 31.8%, respectively) (Table IIC). Although overall *T. cruzi* was identified in similar proportions in seronegative (59.2%) and seropositive (66.7%) samples by combined results from both PCR methods, the prevalence of *T. cruzi* infections identified using epPCR was significantly greater (t = -2.38, df = 34, P = 0.01) in seronegative (75.4%) vs seropositive (45.2%) individuals [Supplementary data (Table V)].

TABLE II

*Trypanosoma cruzi* infection diagnosis of primary diagnosis group (PDG) and secondary diagnosis group (SDG): (A) correlation between serological, end point polymerase chain reaction (epPCR), quantitative polymerase chain reaction (qPCR) and global diagnoses; (B) sensitivity and specificity of conventional serology, epPCR, and qPCR based on the global gold standard; and (C) concordance between serological diagnosis and epPCR, qPCR and global diagnoses, and between primary and secondary results from patients having previous diagnosis (SDG).

(A). Correlation

|  |  | PDG | | | |
|---|---|---|---|---|---|
|  |  | Dx serology | Dx epPCR | Dx qPCR | Dx global |
| SDG | Dx serology | 1 | -0.037 | 0.166 | 0.107 |
|  | Dx epPCR | -0.433 | 1 | **0.313** | **0.649** |
|  | Dx qPCR | 0.143 | **0.389** | 1 | **0.649** |
|  | Dx global | -0.433 | **1.000** | **0.389** | 1 |

(B). Sensitivity and specificity

|  | PDG | | | SDG | | |
|---|---|---|---|---|---|---|
|  | Sensitivity | | | | | |
|  | Mean | 95% CI | | Mean | 95% CI | |
| Dx serology | 0.08 | 0.00 | 0.17 | 0.48 | 0.31 | 0.65 |
| Dx epPCR | 0.84 | 0.72 | 0.96 | 1.00 | 1.00 | 1.00 |
| Dx qPCR | 0.84 | 0.72 | 0.96 | 0.60 | 0.44 | 0.76 |
|  | Specificity | | | | | |
|  | Mean | 95% CI | | Mean | 95% CI | |
| Dx serology | 0.16 | 0.04 | 0.28 | 0.13 | 0.01 | 0.25 |
| Dx epPCR | 0.50 | 0.33 | 0.67 | 1.00 | 1.00 | 1.00 |
| Dx qPCR | 0.50 | 0.33 | 0.67 | 0.60 | 0.44 | 0.76 |

(C). Concordance

|  | PDG | | SDG | |
|---|---|---|---|---|
|  | Dx serology vs | | | |
|  | Concordance | κ (95% CI) | Concordance | κ (95% CI) |
| Dx epPCR | 29.5% (13/44) | 0.08 (-0.58,0.74) | 31.8% (14/44) | 0.94 (0.31,1.57) |
| Dx qPCR | 34.0% (15/44) | -0.36 (-1.03,0.30) | 54.5% (24/44) | -0.28 (-0.88,0.31) |
| Dx global | 20.4% (9/44) | -0.30 (-1.16,0.55) | 31.8% (14/44) | 0.94 (0.31,1.57) |
| Dx primary serology | NA | NA | 36.3% (16/44) | -0.008 (-0.63,0.61) |

*Trypanosoma cruzi SAT haplotypes identified using epPCR* - The SAT gene amplified from 95% of the PDG *T. cruzi* infections and therefore haplotype analysis is representative for all three sites. The SAT gene amplified and sequenced similarly for SDG populations only from SC and SRN sites (86% of infections), but only 19% of infections from SCP, possibly due to this site having the lowest parasite levels measured using qPCR (Tables II-III). Therefore, results for the SDG from SCP should be interpreted with caution [Supplementary data (Table VI)]. A total of 36 SAT haplotypes were identified from 46 consensus sequences (124-126 bp); only 8% (3) were identified in more than one site and similarly from both diagnostic groups. There were significantly more haplotypes from seropositive (91%, N = 11) vs. seronegative (79%, N = 29) infections, although similar prevalence of unique haplotypes (91% vs. 90%, respectively), consistent with high haplotype diversity (Hd = 0.97).

SAT sequence alignments had highest identity for three strains of TcI (Las Palomas, CARI06 and Silvio X10), but most also had high identity for CL Brener (TcVI), strains 3869 and M6241 of TcIII, and strains B147 and 115 of TcV [Supplementary data (Table VII)]. The minimum spanning Network (using the 36 haplo-

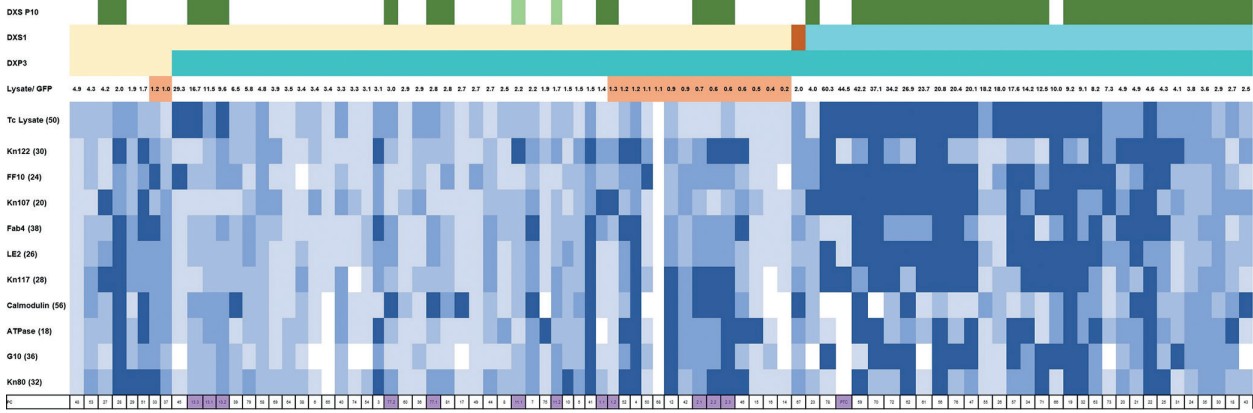

Fig. 2: heatmap of combined results for Oaxaca patient samples reactive to recombinant *Trypanosoma cruzi* proteins: (1) previous conventional serology (DxS Pre10): dk green = 2 test pos., lt green = 1 test pos, no colour = not tested (primary diagnosis group - PDG), (2) current conventional serology (DxS1): yellow = negative to 4 tests, brown = discordant (1 of 4 test pos.), blue = 3-4 of 4 tests pos., (3) combined serological and molecular diagnoses (DxP3): yellow = negative for 4 serological tests, negative 5 genes using end point polymerase chain reaction (epPCR), negative 2 genes using quantitative polymerase chain reaction (qPCR); blue = positive serology (≥ 2 tests positive) and/or epPCR and/or qPCR, (4) Y lysate/GFP ratio: negative = 1.4, < 1.4 in orange, (5) reactivity for Y lysate and 10 recombinant proteins above control (negative + 3SD), and (6) PC - patient codes (numbers with 1-3 decimals in purple indicate sequential samples of individual patients at t = 0 (2016), 12mo, 18mo, or PTC control).

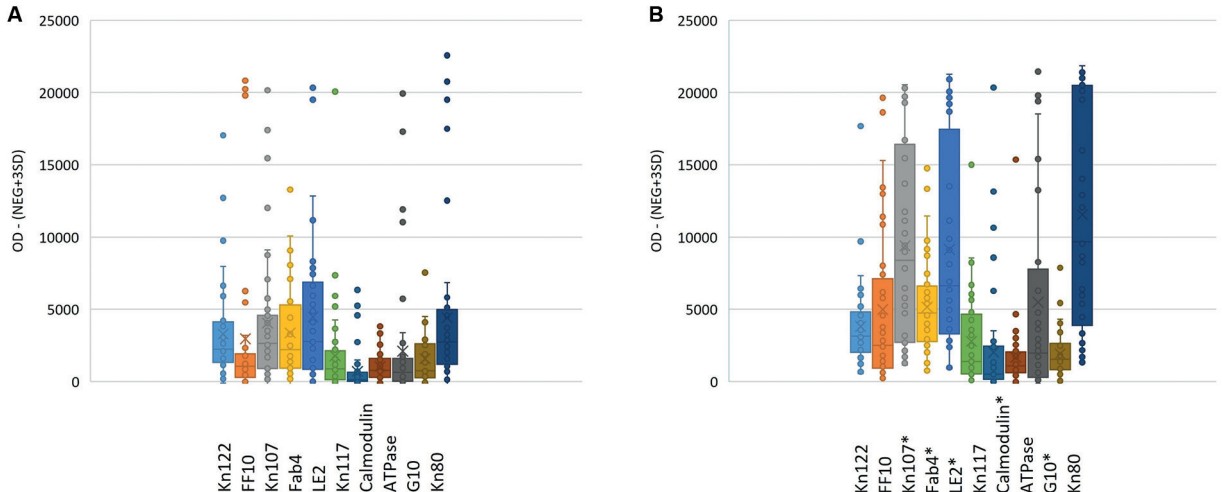

Fig. 3: reactivity of Oaxaca sera to recombinant proteins and Y lysate from (A) primary diagnosis group (PDG) and (B) secondary diagnosis group (SDG). All values represent the optical density (OD) minus (mean negative control + 3SD); all outliers are included. *significant difference between PDG and SDG.

types) evidence at least three shared among the three sites (H2), while one (H6) was shared between SC and SRN, and one (H16) was shared between SCP and SRN. Haplotypes from SRN have more mutational steps from H2, even at the intra-locus level (*e.g.*, between H6 and H32, 18 mutational steps). Similarly, there was at least 18 mutational steps between H2 and H11 from SC (Fig. 4).

Although in far lower overall proportion as compared to SAT, significantly more *T. cruzi* infections were identified using the S34/S67 kDNA/epPCR for the PDG (37.5%; t = -2.02, df = 28, P = 0.02) vs. the SDG (12.0%). A greater proportion of infections were identified using kDNA from SCP as compared to SRN or SC for both the PDG (83.3%, t = -3.44, df = 37, P = 0.0007) and the SDG (66.7%, t = -3.89, df = 33, P = 0.0002). Although the internal kDNA fragment used herein could have pro-

duced a 70-71 bp consensus sequence similar to that for reference samples from the Yucatan (accession numbers OQ236562- OQ236563), final alignments obtained and reported herein from Oaxaca are only 30-34 bp. A total of four haplotypes were identified from 15 kDNA consensus sequences indicating a low to moderate (Hd = 0.54) haplotype diversity. The most frequent haplotype H1 was shared between patients from SCP (9) and SRN (1), while one unique haplotype was identified in each of the sites: SRN (H2, N = 1), SCP (H3, N = 3), and SC (H4, N = 1).

Since there was low amplicon specificity for the ME, 18S and 24S gene sequences, genotype could only be determined using direct sequence identity; amplicon *T. cruzi* specificity for all fragments and for all sites are summarised in Supplementary data (Table III) and for lineage using ME and 24S in Supplementary data (Table

TABLE III

Characteristics of *Trypanosoma cruzi* molecular diagnosis in Oaxacan populations

| County (N) | Previous *T. cruzi* testing (N) | *T. cruzi* only Dx molecular (N) | *T. cruzi* infections epPCR (N) | Proportion *T. cruzi* epPCR ≥ 2g (N) | *T. cruzi* identity SAT | kDNA | ME | 18S | Adjusted parasite load SAT (para/mL) | *T. cruzi* infections qPCR (N) | Adjusted parasite load kDNA (para/mL) |
|---|---|---|---|---|---|---|---|---|---|---|---|
| Salina Cruz (SC, 23) | PDG (13) | 100% (13) | 76.9% (13) | 41.7% (12) | 100.0% | 10.0% | 20.0% | 0.0% | 0.117[b1] | 100.0%[b2] (13) | 0.044[a1,b1] |
|  | SDG (10) | 100% (10) | 100% (10) | 30.0% (10) | 90.0%[a4] | 0.0% | 20.0% | 20.0% | 0.025 | 50.0%[b1] (10) | 0.005[b1] |
| Santos Reyes Nopala (SRN, 28) | PDG (12) | 83.3% (12) | 83.3% (12) | 30.0% (10) | 100.0%[§1] | 10.0% | 10.0%* | 20.0% | 0.145[a1,b1] | 100.0%[b2] (12) | 0.039[a2,b1] |
|  | SDG (16) | 18.8% (16) | 75.0% (16) | 16.7% (12) | 83.3%[§3] | 8.3% | 25.0%[§1] | 0.0% | 0.109 | 68.8% (16) | - |
| Santa Cruz Papalutla (SCP, 39) | PDG (19) | 100% (13) | 63.2% (19) | 91.7% (12) | 100.0% | 83.3% | 0.0% | 5.3% | 0.025[a1] | 36.8%[b5] (19) | 0.016[b5] |
|  | SDG (18) | 0 (18) | 16.7% (18) | 66.7% (3) | 100.0% | 66.7% | 0.0% | 0.0% | 0.079 | 16.7%[b2] (18) | -[b2] |
| Oaxaca (88) | PDG (44) | 94.7% (38) | 72.7% (44) | 53.1% (32) | 100.0%[§1] | 37.5% | 9.4% | 9.4% | 0.103[a2,b2] | 72.7%[b9] (44) | 0.044[a3,b7] |
|  | SDG (44) | 31.0% (44) | 56.8% (44) | 28.0% (25) | 88.0%[§7] | 12.0% | 20.0%[§3] | 8.0% | 0.089 | 43.2%[b3] (44) | 0.011[b3] |

*1 sample amplifying three genes end point polymerase chain reaction (epPCR) + satellite (SAT) quantitative polymerase chain reaction (qPCR); §#: *T. dionisii* 18S co-infection; *a* #: SAT outliers (1488 and 2 p/mL) and kDNA outliers (5, 8, 10 p/mL) not included; *b* #: non-quantitative (NQ) *T. cruzi* also positive by epPCR, serology or qPCR.

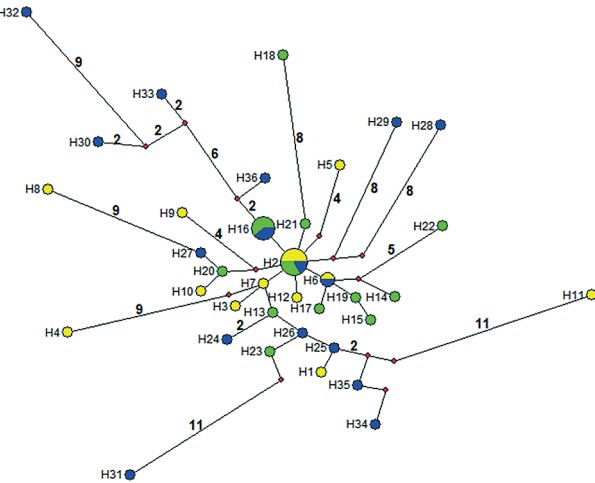

Fig. 4: minimum haplotype Network for *Trypanosoma cruzi* based on 134 nucleotides of the satellite gene fragment (SAT). Haplotype colour represents each site. Green: Santa Cruz Papalutla (SCP), blue: Santos Reyes Nopala (SRN), yellow: Salina Cruz (SC). Missing haplotypes are indicated as red circles. The line connecting haplotypes represents one mutational step, whereas numbers along the lines are total number of mutational steps. Circle area is proportional to the frequency of each haplotype.

VIII). The ME gene was not identified from amplicons of any PDG or SDG (20/29) infection from SCP, in contrast to the greater prevalence of infections identified using kDNA from that site. No ME-LII amplicons (300 bp) had identity to *T. cruzi*, while 15% of samples amplifying single ME-LI amplicons and 24% of samples amplifying both LI and LII amplicons had identity to TcI. A total of eight ME haplotypes (Hd = 1.00) were identified, three from SC and five from SRN. A total of 63.3% of samples amplifying ME for either LI and/or LII also amplified 24S, although none of the sequences from 24S amplicons, ranging between 110 bp and 130 bp had identity with *T. cruzi*.

The 18S gene (561 bp) amplified in 36% of *T. cruzi* infected samples producing amplicons between 550-650 bp. Consensus sequences (541 bp) of four samples had high query cover (range from 93%-100%) and identity (range from 81%-99%) to *T. cruzi* for TcI (Strain G) from triatomines (*T. barberi*, *T. dimidiata*), dogs, birds and both North and South American marsupials and rodents. Although few samples were analysed, the 18S *T. cruzi* haplotype diversity was high (Hd = 1.0), with two haplotypes from SC (H1 and H2) and two from SRN (H3 and H4).

Most samples amplifying the 18S (70%) produced multiple bands within the 350-550 bp range which had identity to *Trypanosoma dionisii*. *T. dionisii* 18S sequences ranged from 347-504 bp, with high query cover (range from 96%-100%) and identity (range from 92%-100%) among the first 100 matches. A total of five *T. dionisii* 18S haplotypes were obtained from seven patient' samples; haplotype diversity was high (Hd = 0.85). Only one haplotype (H1) was shared between two patients, one each from SC and SRN, and additional unique haplotypes H2-H5 were identified only from SRN; no *T. dionisii* was identified from SCP. All *T. dionisii* infections

(N = 7) were identified only in *T. cruzi* co-infected samples of the PDG and the SDG from SRN and the SDG from SC. Four of seven *T. cruzi* co-infected samples were seropositive using conventional *T. cruzi* serology, while all seven co-infections were confirmed for *T. cruzi* using SAT and/or ME epPCR and SATqPCR.

*TaqMan qPCR (SAT and kDNA) and parasite load quantification - Trypanosoma cruzi* infection prevalence using qPCR (73% for the PDG and 43% for the SDG) was similar to that using epPCR overall (Table I), although there were *T. cruzi* prevalence differences for the PDG among sites between the two PCR methods. The proportion of infections from SCP identified using qPCR was significantly lower for both PDG (37%, t = -2.31, df = 28, P = 0.0001) and SDG (17%, t = -1.28, df = 33, P = 0.0003) diagnostic groups, as compared to SRN (100% and 69%, respectively) and SC (100% and 50%, respectively). *T. cruzi* infection using qPCR correlated 65% with global diagnosis for the PDG (47.8% for SAT), but only 39% for the SDG (46.8% for SAT) [Table II, Supplementary data (Table IV)]. The correlation of qPCR with serological diagnosis was low but similar in both diagnostic groups (17% in PDG and 14% in SDG), as was concordance for the PDG (34.0%, K = -0.36, 95% CI: -1.03, 0.30) and the SDG (54.5%, K = -0.28, 95% CI: -0.88, 0.31) (Table II). The sensitivity of qPCR was 84% for the PDG and 60% for the SDG and its specificity was 50% for the PDG and 60% for the SDG. Although there was similar *T. cruzi* infection prevalence using both SAT and kDNA genes by qPCR in seronegative (61.4%) and seropositive (51.6%) samples, the proportion of non-quantifiable infections using kDNA (31.3%) was significantly greater than that using SAT (5.0%; t = -2.64, df = 108, P = 0.004) [Supplementary data (Table V)].

Parasite load values from all samples were normalised based on Ct values from spiked titration series for SAT and kDNA [Table III, Supplementary data (Fig. 2)]. Overall, median parasite loads for the PDG and the SDG were low using both the SAT (0.103 equivalent parasites/mL and 0.089 equivalent parasites/mL, respectively) and the kDNA (0.044 equivalent parasites/mL and 0.011 equivalent parasites/mL, respectively). High parasite load outliers, not included in median load values, were only recorded from the PDG (10% of SAT and 20% of kDNA) from all three sites (Table III). High SAT outliers were recorded from SCP and SRN (1488 and 2 para/mL, respectively) and kDNA outliers from SC and SRN (5, 8, and 10 para/mL).

## DISCUSSION

Only 81% of *T. cruzi* infections diagnosed 10-15 years previously in three separate qualified laboratories were serologically reconfirmed, surprising since 14% additional infections were confirmed using PCR. There was even less capacity to diagnose more recent infections (PDG) with only 8% sensitivity for serology, while 84% for PCR, suggesting limited capacity of antigens/epitopes used in current commercial tests to detect predominant antibody responses to *T. cruzi* populations. Reactivity to recombinant expression proteins

validated the presence of *T. cruzi*-specific antibodies in both subgroups, significantly greater for Kn122, FF10, Kn80, Kn107, Fab4, LE2, and support results for the failure of conventional and rapid diagnostic tests. Specificity of conventional serology, the capacity to identify true negatives, was similar for both diagnostic-history groups (13% - 16%) indicating that both conventional and rapid tests are additionally ineffective as diagnostic tools to determine the absence of *T. cruzi* infection in exposed populations. The evidence indicates a shift over the course of infections in antibody responses to current commercial test antigens/ recombinants, at least across the Zapotecan and Mixtecan regions, since these results have been substantiated using additional commercial serological tests and through immunomic analyses.[32,33]

Multiple international studies have evaluated standardised *T. cruzi* diagnostic serology, since they can provide timely and highly cost-effective tools to detect and initiate treatment to prevent CD burden, particularly if sensitive and specific in rapid format for point of care tests.[58] However, the mounting robust evidence for heterogeneous human immunogenetics and immune responses across the American continent,[59,60] and multiple studies demonstrating evolving immune responses to pathogen population shifts due to anthropogenic landscape-biodiversity changes have not been evaluated for *T. cruzi*.[61,62] Consistent with increasing evidence for reduced sensitivity and specificity of current "universal" conventional and rapid tests in some regions,[63,64] these results are not surprising given the parasite's high genetic diversity as documented from vectors and reservoirs from these Oaxacan sites.[40] Failure to include discordant sera and relevant regional and disease-chronicity antigens in control panels of international validations have led to performance measure bias, at least for Mexico. This bias has formed the basis for Mexican PHS norms and policy, "overriding" the wealth of evidence from the present and other detailed studies and antigen-antibody specificities in Mexican populations.[32,33] Independent of the inflexible public policy, there is an urgent need to design and assure access to a new diagnostic algorithm for the estimated 85,000 incident annual cases in Mexico, and probably the North American and Mesoamerican regions.[8] Given low negative predictive value for current commercial serological tests and risk for congenital, transfusion and transplant transmission, this would require, in the short-term, testing seronegative samples using standardised and validated in-house SAT qPCR and amplifying testing capacity to clinical laboratories with molecular capacity, at least until commercial qPCR improve sensitivity.

Lack of concordance between serological and molecular diagnosis reflects the methodological capacity of each, the former dependent on the measured phenotype specificity and the latter to the balance of circulating parasite levels and global antibody responses limiting those levels. Naturally shifting circulating parasites of varying antigenic diversity are the true measure of active *T. cruzi* infection, unless there is more precise capacity to detect chronic specificities of immune responses. In the present study, molecular diagnostic methods were fundamental to reconfirm a fifth of the SDG and most of the PDG

infections, not confirmed using conventional serological tests. Therefore, lack of concordance between serological and molecular tests evidence methodological diagnostic failures, which justify the need to use both in a complementary way until they can be improved and individually validated for precise measure of *T. cruzi* infection considering origin, history of infection and disease status. The overall proportion of both epPCR and qPCR positives among seropositive individuals (58%) was lower in SDG infections, very possibly due to the lower parasite loads, barely at detection threshold levels. In contrast, a similar proportion of molecular positive infections identified in "apparent seronegative" individuals (86%) evidence that antibodies not identified by current serological tests may be specific to regional or disease history novel antigens/epitopes.[32,33] All evidence supports the need to improve specificity of overall infection diagnosis given the low negative predictive value of current serology and additional benefit for timely diagnosis (and treatment), since undiagnosed, misdiagnosed or diagnosed and untreated patients are a continued source for chronic disease outcomes, CD burden, vertical transmission as well as vector infections.[3,6] Diagnostic policy in Mexico must consider the estimated 6-8 million current human *T. cruzi* infections which are evolving and re-infecting with alternative parasite populations, since greater than 98% have never been diagnosed or treated.[7,8]

However, in the medium-term, public policy should also target the design and testing of new serological tests using the growing evidence for appropriate Mexican-relevant antigens,[32,33,65] since governance and budgetary considerations have consistently been important access hurdles for CD in Mexico. Any new formulations will require broad validation across other Mexican regions (occidental, northwest, northeast, Gulf of Mexico), since serological diagnostic failure has been heterogeneous even within this reduced region (4.8% of landmass and 3.3% of the population).[66] This failure affected primary diagnoses (recent infections) in an urban area with profound landscape modifications (SC) and in a highly modified rural landscape (SCP), despite reconfirming fewer infections from rural highly modified landscape (SRN). Serological diagnostic performance is expected to be even more heterogeneous than measured herein across Mexico and will require using the global gold standard and including acute and chronic cohorts from rural and urban demographics as well as clinically important CD classification cases (asymptomatic, symptomatic). Since the performance (sensitivity, specificity) of molecular methods for Mexican *T. cruzi* infections and populations confirmed patent parasitaemia in conventional test seronegatives and seropositives, the evaluation study design will need to include serological and molecular testing for all participants and diverse population subgroups based on demographics, clinical disease and all transmission-relevant ecoregions.

*Trypanosoma cruzi* infection, confirmed by sequence identity, was sensitive only for SAT amplicons in this study. SAT haplotype diversity was higher in recent PDG seropositive infections (vs SDG), although the proportion of unique haplotypes was similar between the two groups and among sites. There was high interpopulation SAT haplotype diversity since only one of the 40 was shared among all three sites and only two were shared each by two sites (SC-SRN and SRN-SCP). Haplotypes from SCP, the most urban, modified and transport-connected landscape in the Neovolcanic central valleys (*T. barberi*) were closer to the site-common haplotype than to the two coastal sites, as expected. We found no evidence of genotype associations with SAT haplotypes in these Mexican populations which had highest identity to TcI (Silvio x10 and CARI-06), although lower but high identity also to lineages TcIII, TcV, and TcVI, indicating that SAT repeats rich in AT are conserved across lineages.[67,68]

Previous studies in Mexico using the S35/S36 fragment of kDNA with epPCR reported 41% amplification from asymptomatic seropositive blood donors[69] and 81% amplification from seropositive CCC cases,[70] although neither study reported sequence specificity of amplicons. Herein, the internal shorter kDNA identified only 26% of overall infections, 93% of these were also identified using either SAT or ME. Differential success of the internal shorter kDNA sequence may be due to the complementary of the primer 5' region being located in a variable region, which would also explain the low efficiency to sequence from kDNA amplicons.[40,50] Only four haplotypes were identified from 15 kDNA sequences, the only shared haplotype identified from SRN and SCP. The S34/S67 fragment provides evidence for a broader kDNA diversity in human Oaxacan populations, although novel kDNA haplotypes having low prevalence in zoonotic reservoirs are present in this region.[71]

Very low patent parasitaemia and/or sequence polymorphisms affected results for the three gene markers used to genotype parasite populations (ME, 24S, 18S). Although 77.5% of infected samples amplified for the ME lineage I and/or lineage II, only 13% of these had sufficient aligned nucleotides to score identity for *T. cruzi* DTUI. Buekens et al. previously reported genotyping only 41% of parasite populations in pregnant Mexican women and García et al. reported similar results for another ME fragment and ribosomal markers.[72,73] Although novel methods provide new opportunities to detect and genotype parasite populations,[74] if current oligonucleotides for the major ME lineages are used to filter NGS analyses from human infections to analyse *T. cruzi* diversity, this study's evidence suggests results will be biased.

Unexpected co-infections of *T. cruzi* and *T. dionisii* were identified in two of the Oaxacan sites, both along or within the foothills of the Pacific coast. *T. dionisii* was identified in four *T. cruzi* serology-positive and three seronegative individuals (the latter were SAT epPCR and/or qPCR positive). Given evidence of *T. dionisii* in triatomines, bats and spillover to dogs and synanthropic rodents in these communities,[40] simultaneous vector transmission of both *T. cruzi* and *T. dionisii* is likely to be occurring. How this may affect human population immune responses and those specific to *T. cruzi*, as well as performance of diagnostic tests, will require further study of individual *T. dionisii* infections.

If CD is to be appropriately targeted at all healthcare levels in Mexico, it will require access to sensitive as well as specific diagnosis for *T. cruzi* infection. The Mexican National Institute for Epidemiological Diagnosis and Reference (InDRE) tests all proposed commercial serological kits for *T. cruzi* (ELISA, quimioluminescence, rapid tests, multiple antigen types) using in-house characterised reference and/or commercial reference panels approved for national and state procurement.[4] However, the use of antibody as a proxy for infection depends on the capacity to recognise a broad repertoire of immune responses (or adequate evidence of conserved responses) to an even broader parasite diversity, both affected by reservoir and vector diversity and parasite population shifts over the chronicity of infections. [75] Current control panels do not include discordant or seronegative samples which are *T. cruzi* molecular positives and there has been no evaluation or analysis of control panel sera representation for antibody responses to *T. cruzi* across the heterogeneous epidemiological gradient currently evident in Mexico.[32,40] If only seropositive samples to "apparently" universal commercial tests are used, as opposed to a full panel of control sera for the high diversity and range of circulating parasite populations and antibody responses in Mexican populations, serology for *T. cruzi* infection will continue to be insensitive, imprecise and ineffective for precise and reliable diagnosis. Continued neglect to provide accurate infection diagnosis maintains social inequity hindering public policy change and civil society efforts to implement an integrated approach for CD case detection and management in Mexico.[8]

## ACKNOWLEDGEMENTS

To Dr Rick Tarleton for support and analysis of samples using Luminex, and Dr Guillermo Ruiz Palacios for logistical support of INCMNSZ personnel. We would also like to thank technical field and insectary personnel Antonio Lopez Morales, community outreach project support from Adriana Cruz-Celis, Gonzalo de la Parra and Daniel Vazquez, sample acquisition and processing from José Asunción Nettel Cruz, and Nora Mora-Suarez, as well as all community and civil authority support and collaboration at all sites participating in this study.

## AUTHORS' CONTRIBUTION

Conceptualisation - JMR and GSG; methodology - JMR, KCF, APM, MVC, KEL and MRR; software - JMR, KCF and APM; validation - JMR, KCF, APM, ETK, KEL and AGS; formal analysis - ETK, SG, APM and KEL; investigation - JMR, KCF, ETK, MVC and APM; data curation - JMR, KCF, MVC and APM; writing - original draft preparation - JMR, APM and GSG; writing - review & editing - JMR, APM, GSG and AGS; project administration - JMR and KCF; funding acquisition - JMR and AGS. All authors have read and approved the final manuscript. The authors declare that they have no conflicts of interest.

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

# OPEN PEER REVIEW

Memórias do IOC thanks the anonymous reviewers for their contribution to the peer review of this work.

**FIRST REVIEW ROUND**

REVIEWERS COMMENTS

### REVIEWER #1

The manuscript presents a compelling investigation into the diagnostic challenges of Trypanosoma cruzi infection in Mexican populations, with a focus on evaluating the sensitivity and specificity of conventional serological and molecular diagnostic methods. The study contributes insights into the limitations of current diagnostics and proposes a global gold standard approach. The data are well-structured and thoroughly analyzed, which enhances its relevance for advancing diagnostic methodologies for Chagas disease.

As strength, the manuscript is well-structured and methodologically robust, offering a comprehensive evaluation of diagnostic challenges in T. cruzi infection through the integration of serological and molecular methods, while addressing a critical public health issue with detailed statistical analyses and relevant population focus. Despite these strengths, there are opportunities for improvement and limitations that should be addressed before being accepted for publication.

Abstract and Introduction:

1. The abstract lacks clarity in describing the main conclusions and practical implications of the findings. Refining this section to emphasize the diagnostic challenges and proposed solutions would enhance its impact.

2. The introduction is comprehensive but could benefit from a clearer articulation of the study's hypothesis or research gap.

Data Presentation:

1. Some tables and figures (e.g., heatmaps, haplotype networks) require better integration into the main text for contextual understanding.

2. Supplementary data references are frequent but insufficiently described, requiring readers to constantly consult external materials.

Results Interpretation:

1. The low concordance between serological and molecular methods needs a more detailed discussion. Exploring the potential causes, such as parasite strain diversity or immune response variability, would add depth to the conclusions.

2. The implications of co-infections with T. dionisii should be further explored, particularly in the context of cross-reactivity or diagnostic interference.

Language and Style:

1. Some sections contain complex sentence structures and technical jargon that may hinder accessibility for non-specialist readers. Simplifying language without losing scientific accuracy would improve readability.

2. Inconsistent use of abbreviations and terms (e.g., PDG, SDG, SAT, kDNA) detracts from clarity.

Practical Implications and Recommendations:

1. While the manuscript identifies diagnostic shortcomings, it provides limited actionable recommendations for improving diagnostic protocols in resource-limited settings.

2. Discussing the feasibility of implementing the "global gold standard" approach in endemic areas would enhance the translational value of the findings.

To my mind, the manuscript is a significant contribution to Chagas disease diagnostics. Addressing the identified weaknesses and incorporating the suggested improvements would enhance its clarity, impact, and utility for both researchers and public health practitioners. I recommend conducting a thorough language edit to ensure consistency, reduce redundancy, and improve flow. Finally, I recommend its acceptance with major revisions.

### REVIEWER #2

This manuscript examines the sensitivity and specificity of current serological and molecular diagnostic methods for Chagas disease in populations from three regions of Oaxaca, Mexico. The proposal is particularly relevant in this region due to the discrepancies in diagnostic results reported in previous studies.

Participants were selected based on clinical and epidemiological criteria (Primary Diagnosis Group, PDG) and 45 previously diagnosed patients who had not received treatment (Secondary Diagnosis Group, SDG).

It is unclear whether samples from individuals known to be negative for Chagas disease were used to monitor cross-contamination during the DNA extraction and amplification process and to assess the specificity of the PCR reactions.

However, the study's design posed challenges. It involved administering a large number of diagnostic tests, both molecular and serological, and not all samples were evaluated using every method. Consequently, analyzing the data and interpreting the results effectively took much work.

Which method was considered gold standard?

It is crucial to clearly define the specific type of data used in the statistical calculations. For example:

1. "Pearson's correlation coefficient measures the strength and direction of the linear relationship between two quantitative variables that follow a parametric distribution." It is necessary to clarify which values from the ELISA and qPCR tests were utilized for this analysis. Additionally, the sample size (N) must be specified, along with a clear definition of the data considered to be the global standard.

2. Specificity, defined as the ability of a diagnostic test to accurately identify individuals who do not have the disease, also requires clarification. How was the specificity of the PCR assays calculated? Were samples from patients with other diseases, such as leishmaniasis or Chagas disease negative, included in this assessment?

## AUTHORS' RESPONSE TO THE REVIEWERS

Responses to the reviewers´ comments on the manuscript ID MIOC-2024-0224 entitled "Diagnosis of Trypanosoma cruzi infection in Mexican populations: current conventional serology lacks adequate sensitivity and specificity". All authors are grateful for their observations, suggestions and comments to improve the manuscript.

### REVIEWER #1

Abstract and Introduction:

1. The abstract lacks clarity in describing the main conclusions and practical implications of the findings. Refining this section to emphasize the diagnostic challenges and proposed solutions would enhance its impact.

Response: the section has been edited and findings-challenges and primary solutions highlighted

2. The introduction is comprehensive but could benefit from a clearer articulation of the study's hypothesis or research gap.

Response: The section has been edited and the research gap as well as study hypothesis more specifically highlighted.

Data Presentation:

1. Some tables and figures (e.g., heatmaps, haplotype networks) require better integration into the main text for contextual understanding.

Response: The two figures mentioned are fully described and integrated now into the main text so that contextual understanding is clearer. We have reviewed and expanded all other tables and figures as well.

2. Supplementary data references are frequent but insufficiently described, requiring readers to constantly consult external materials.

Response: We have reviewed and edited all references in the main text referring to data detailed in the supplementary tables so that readers would only need to review these latter tables if they wish to check additional data details. Due to editorial limitations on text length, full description of all data would have been unacceptable, but given the surprising data generated, we felt necessary to make these available and transparent for our colleagues.

Results Interpretation:

1. The low concordance between serological and molecular methods needs a more detailed discussion. Exploring the potential causes, such as parasite strain diversity or immune response variability, would add depth to the conclusions.

Response: We have amplified the discussion regarding concordance, precisely related to and using multiple details summarized in the supplementary tables (which could not be adequately highlighted or discussed due to manuscript length limitations). The failure and limitations of both conventional serology (limited/imprecise immune response detection) and molecular markers/PCR sensitivity are further mentioned in this issue.

2. The implications of co-infections with T. dionisii should be further explored, particularly in the context of cross-reactivity or diagnostic interference.

Response: We have edited and further mentioned the data we have for the few cases of con-infections, although without having individual infections of T. dionisii in humans and being able to analyze IR to specific antigens or proteins, it is impossible with this study to generate further hypotheses. A specific study in these populations would need to determine if in fact any disease symptoms result from the dionisii infection, or if is affects T cruzi specific serological responses.

Language and Style:

1. Some sections contain complex sentence structures and technical jargon that may hinder accessibility for non-specialist readers. Simplifying language without losing scientific accuracy would improve readability.

Response: We have fully re-edited the text and reduced to the minimum the use of technical language as long as it does not favor imprecise or generalized reporting of results, or of data interpretation and discussion.

2. Inconsistent use of abbreviations and terms (e.g., PDG, SDG, SAT, kDNA) detracts from clarity.

Response: We have attempted to be more precise in the editing and new version

Practical Implications and Recommendations:

1. While the manuscript identifies diagnostic shortcomings, it provides limited actionable recommendations for improving diagnostic protocols in resource-limited settings.

Response: We have presented two recommendations for the short and medium term for the failure of serology: resolve seronegatives using PCR, and develop-test new multiantigen formulations for serological tests.

Mexico is not a resource-limited country and even first level healthcare for many infectious pathogens or indeed diseases (ie. Tuberculosis) have not had limited attention, as is the case of CD. The prevalence is daunting and as it is a chronic disease, it is too easy to sweep under the PHS attention.

2. Discussing the feasibility of implementing the "global gold standard" approach in endemic areas would enhance the translational value of the findings.

Response: We have edited the discussion and attempted to present this need, particularly for clinical settings and with their broadening capacity for molecular diagnoses. In fact, clinicians are those with increasing requests to test seronegative patients for T cruzi.

## REVIEWER #2

Which method was considered gold standard?

Response: this is specifically defined in Methods lines 253 to 256, and just prior to this, specific description of serological, epPCR and qPCR diagnoses.

It is crucial to clearly define the specific type of data used in the statistical calculations. For example:

1. "Pearson's correlation coefficient measures the strength and direction of the linear relationship between two quantitative variables that follow a parametric distribution." It is necessary to clarify which values from the ELISA and qPCR tests were utilized for this analysis.

Response: in Methods, lines 247-252 specifically details that we are analyzing as positive or negative for each specific test (each serological test, epPCR or qPCR), for each sample and according to their assigned cohort. lines 257-265 details which correlations for these same results were calculated. Additionally, the sample size (N) must be specified, along with a clear definition of the data considered to be the global standard.

Response: All tables, primary and supplemental, give al sample sizes (N) for each group.

2. Specificity, defined as the ability of a diagnostic test to accurately identify individuals who do not have the disease, also requires clarification.

Response: in Methods line 266: True negative/(true negatives + false positives).

How was the specificity of the PCR assays calculated?

Response: specificity for PCR was calculated in the same way as any other, ie. True negative/(true negatives + false positives).

Were samples from patients with other diseases, such as leishmaniasis or Chagas disease negative, included in this assessment?

Response: yes, we have a panel of 20 negative samples analyzed for all tests used in parallel to control for all methods.

## SECOND REVIEW ROUND

REVIEWERS COMMENTS

## REVIEWER #1

Thank you for the opportunity to review this manuscript. The study addresses a critical diagnostic challenge in Chagas disease endemic to Mexico and highlights valuable data on the inadequate performance of conventional serological assays, while also supporting the utility of molecular methods. The work is relevant to both clinical and public health contexts. However, some points should be addressed.

Major Limitations

1. The use of a combined "global gold standard" (≥2 serological positives and/or PCR positives) increases diagnostic sensitivity but risks inflating true positivity, particularly in settings where molecular positivity may reflect transient parasitemia or false positives. This definition is operational rather than validated and may bias performance estimates of individual tests.

2. Despite thorough diagnostic analyses, the study does not correlate test results with clinical outcomes (e.g., ECG abnormalities, symptom profiles). This makes it difficult to assess the implications of seronegative-PCR-positive cases in terms of disease staging or need for treatment.

3. Cohort Characteristics Across Sites are unbalanced. Indeed, differences in age, sex ratio, and schooling

(especially in the SRN site) may confound immune response patterns and diagnostic performance but are not statistically controlled. This weakens the strength of comparisons across populations and sites.

4. While the study demonstrates low specificity in serological tests, it does not sufficiently explore the causes (such as cross-reactivity with T. rangeli, T. dionisii), or other trypanosomatids. The role of non-specific immune responses or background reactivity is also not fully addressed.

5. Although Luminex-based profiling of recombinant proteins was performed, the manuscript does not fully capitalize on this dataset to identify potential antigen combinations or patterns that might improve future diagnostic test development for local T. cruzi strains.

Minor Limitations

1. The specificity of PCR assays is reported using 20 negative samples, but it remains unclear whether samples from individuals with other parasitic diseases (e.g., leishmaniasis) were included to rigorously test for cross-reactivity.

2. Some abbreviations (e.g., PDG, SDG, SAT, ME) are used inconsistently or without initial clarification, which may reduce clarity for non-specialist readers.

3. Important findings are often relegated to supplementary tables and figures. This disperses the narrative and makes it difficult to fully interpret key outcomes without cross-referencing.

4. The manuscript does not adequately address the feasibility or cost-effectiveness of implementing PCR-based diagnostics in routine surveillance or primary care settings in Mexico.

In my oppinion, this study offers compelling evidence for the reevaluation of current serodiagnostic strategies for Chagas disease in Mexico. The authors are commended for the scope and rigor of their analyses. Addressing the limitations above would strengthen the manuscript's impact and practical relevance.

## AUTHORS' RESPONSE TO THE REVIEWERS

Responses to the reviewers´ comments on the manuscript ID MIOC-2024-0224.R1 entitled "Diagnosis of Trypanosoma cruzi infection in Mexican populations: current conventional serology lacks adequate sensitivity and specificity". All authors are grateful for their observations, suggestions and comments to improve the manuscript.

### REVIEWER #1

Thank you for the opportunity to review this manuscript. The study addresses a critical diagnostic challenge in Chagas disease endemic to Mexico and highlights valuable data on the inadequate performance of conventional serological assays, while also supporting the utility of molecular methods. The work is relevant to both clinical and public health contexts. However, some points should be addressed. Note: We thank the Reviewer for comments and respond here to each point raised as a major or minor limitation; we would like to highlight that none of the major points are considered within the scope of this study´s design and goals to warrant modifying the text. Concerning minor limitations, we have addressed each point and have made any additional changes where necessary with change control in the manuscript.

Major Limitations 1. The use of a combined "global gold standard" (≥2 serological positives and/or PCR positives) increases diagnostic sensitivity but risks inflating true positivity, particularly in settings where molecular positivity may reflect transient parasitemia or false positives. This definition is operational rather than validated and may bias performance estimates of individual tests.

Response: We disagree with the reviewer´s assertion, since we have consistently used the goal of "T. cruzi infection" which includes "transient parasitemia" (the goal is not to detect infectiveness) and for which it is well known that false positives may more probably measure minor cases of false infections due to immunoreactivity of autoantibodies vs. active infection. If immune response reactivity is not sensitive as we now know, the only potential method to complement serology, unless there is an improvement in antigens used for serology, is direct parasite detection, hence our use of PCR as the most sensitive current method.

2. Despite thorough diagnostic analyses, the study does not correlate test results with clinical outcomes (e.g., ECG abnormalities, symptom profiles). This makes it difficult to assess the implications of seronegative-PCR-positive cases in terms of disease staging or need for treatment.

Response: Correlation with clinical outcomes implies a longitudinal analysis, which was not the goal for this cohort of the overall study. We are currently writing a manuscript with a cohort of patients from Yucatan (southeast Mexico) in which the results show no correlation for seronegativity/PCR positive and cardiac manifestations compatible with Chagas disease in the participants, but this is a separate manuscript with different objectives than the one presented here.

3. Cohort Characteristics Across Sites are unbalanced. Indeed, differences in age, sex ratio, and schooling (especially in the SRN site) may confound immune response patterns and diagnostic performance but are not statistically controlled. This weakens the strength of comparisons across populations and sites.

Response: Although age and sex ratio averages are higher and broader for the SRN, an important consideration to validate preliminary evidence with further analyses of larger and better matched cohorts/groups included in this study. However, subgroups within the cohorts (PDG, SDG), were too small to statistically control and therefore we

have taken care not to emphasize or conclude regarding apparent differences in diagnostic precision, and have emphasized within site differences (PDG vs. SDG) and overall combined evidence, and not that across populations and sites.

4. While the study demonstrates low specificity in serological tests, it does not sufficiently explore the causes (such as cross-reactivity with T. rangeli, T. dionisii), or other trypanosomatids. The role of non-specific immune responses or background reactivity is also not fully addressed.

Response: With respect, the presence of T. rangeli and T. dionisii in triatomines and wildlife from these sites has been previously reported (Izeta-Alberdi et al. 2016), species specific and taxonomic landscape evidence will be published soon; the former species has not been identified and the latter has, coinciding with co-infections reported herein. Non-specific immune responses to trypansomatids assumes cross-over proteomes and potential serological sensitivity bias, not analysed in any study published related to serological diagnostic performance to date and was not contemplated as an objective in the present study.

5. Although Luminex-based profiling of recombinant proteins was performed, the manuscript does not fully capitalize on this dataset to identify potential antigen combinations or patterns that might improve future diagnostic test development for local T. cruzi strains.

Response: Evidence that has been cited and published regarding the Luminex RPs cautions regarding specific molecule reactivity, which is the reason we only analyse and highlight differences among the proteins related to serological and PCR reactivity of subgroups based on infection history. We are working collaboratively to improve diagnostic tests as published in Romer et al, Ricci et al and more recently Ossowski et al. has already provided evidence of multiple peptide combinations for improved diagnostic performance.

Minor Limitations 1. The specificity of PCR assays is reported using 20 negative samples, but it remains unclear whether samples from individuals with other parasitic diseases (e.g., leishmaniasis) were included to rigorously test for cross-reactivity.

Response: The primers used to amplify sample DNA have been previously tested on spiked or patient samples having similar phylogenetic organisms (Leishmania, other Trypanosomas – few of which have voucher sequences in GenBank) in original and many subsequent studies (not possible to cite all in references). It was not deemed necessary to include additional non-T cruzi patient samples in the study design either for serology or for PCR, and no sequences with greater than 70% identity and high query cover were identified from any taxonomic proximity to Trypanosoma, none of them blasted for another species.

2. Some abbreviations (e.g., PDG, SDG, SAT, ME) are used inconsistently or without initial clarification, which may reduce clarity for non-specialist readers.

Response: All abbreviations have been checked and first usage details are included as an addendum to these responses. We have modified certain primary citations in the text to assure all first citations of abbreviations include the complete term. These changes have been highlighted in yellow in the manuscript.

3. Important findings are often relegated to supplementary tables and figures. This disperses the narrative and makes it difficult to fully interpret key outcomes without cross-referencing.

Response: Only key evidence and summarized data are included in primary tables and figures due to limitations by the journal for the number of tables and figures. However, we have included disaggregated and details of data not normally published in other studies for full data availability and transparency.

4. The manuscript does not adequately address the feasibility or cost-effectiveness of implementing PCR-based diagnostics in routine surveillance or primary care settings in Mexico.

Response: The goal of the current study was to generate evidence regarding serological performance for T. cruzi diagnosis in Mexican patients and exposed populations. Unfortunately, restrictions on length and content of the manuscript restrict including cost-benefit analyses for different diagnostic protocols vs false serological negativity, although once we have completed publication of the second cohort from another region, we will be exploring overall recommendations including current performance failure and new diagnostic strategies for cost analyses.

List of first citations of Abbreviations: All changes have been highlighted in yellow to facilitate the reviewer's correction of the new version of the manuscript. In Summary epPCR and qPCR are used without giving full name: is it necessary? Chagas disease in humans (CD) line 65 Mini-exon (ME) line 106 end point polymerase chain reaction (epPCR) line 108 satellite (SAT) line 109 18S ribosomal DNA (18S rDNA) line 109 discrete typing unit I (DTUI) line 112 (secondary diagnosis group, SDG) line 130 (primary diagnosis group, PDG) line 131-132 quantitative PCR (qPCR) line 135 PHS (Public Health Service) line 149-150 Sanitary Jurisdiction Heads (JSI, JSII, JSIV line 151 Santos Reyes Nopala (SRN) line 156 Salina Cruz (SC line 157 Sta Cruz Papalutla (SCP line 158 serum separation tube-SST Line 177 guanidine – EDTA buffer (GEB) Line 178 Instituto Nacional de Nutrición y Ciencias Médicas Salvador Zubirán (INNCMSZ) line 182 enzyme-linked immunosorbent assay (ELISA) line 187 optical density (OD) line 188 standard deviation (SD) Line 189 green fluorescent protein (GFP line 197-198 T. cruzi lineage I; TcI line 202 T. cruzi lineage VI; TcVI line 202-203 Instituto de Investigaciones en Ingeniería Genética y Biología Molecular (INGEBI) line 207-208 kinetoplast DNA (kDNA) line 216 24Sα ribosomal DNA (24S rDNA) line 219-220 cytochrome b (cyt b) line 225 threshold counts (Ct) line 249 quantitative values (Q), line 255 non-quantitative (NQ) line 256 Triatoma phyllosoma: T. phyllosoma line 313-314 Triatoma mazzottii ; T. mazzottii line 313 Triatoma dimidiata Hg2: T. dimidiata Hg2 line 314 Triatoma barberi: T. barberi line 315-316 parasites/ml; para/ml) line 319-320 ME lineage I (ME LI) line 338 ME lineage II (ME LII line 339-340 Trypanosoma dionisii. T. dionisii line 486.

## THIRD REVIEW ROUND

REVIEWERS' COMMENTS

**REVIEWER #1**

Reviewer comments:

Dear Editor,

I have carefully reviewed the revised version of the manuscript "Diagnosis of Trypanosoma cruzi infection in Mexican populations: current conventional serology lacks adequate sensitivity and specificity." I am satisfied with the authors' responses to the previous comments and concerns. The revisions have addressed all major issues, and the manuscript now meets the scientific and editorial standards required by the journal. In its current form, I recommend acceptance for publication.

