## [Reviewer Report · OPEN PEER REVIEW - Memórias do IOC thanks the anonymous reviewers for
their contribution to the peer review of this work]

FIRST REVIEW ROUND

REVIEWERS COMMENTS

REVIEWER #1

The manuscript presents a compelling investigation into the diagnostic challenges of
Trypanosoma cruzi infection in Mexican populations, with a focus on evaluating the
sensitivity and specificity of conventional serological and molecular diagnostic
methods. The study contributes insights into the limitations of current diagnostics
and proposes a global gold standard approach. The data are well-structured and
thoroughly analyzed, which enhances its relevance for advancing diagnostic
methodologies for Chagas disease.

As strength, the manuscript is well-structured and methodologically robust, offering
a comprehensive evaluation of diagnostic challenges in T. cruzi infection through
the integration of serological and molecular methods, while addressing a critical
public health issue with detailed statistical analyses and relevant population
focus. Despite these strengths, there are opportunities for improvement and
limitations that should be addressed before being accepted for publication.

Abstract and Introduction:

1. The abstract lacks clarity in describing the main conclusions and practical
implications of the findings. Refining this section to emphasize the diagnostic
challenges and proposed solutions would enhance its impact.

2. The introduction is comprehensive but could benefit from a clearer articulation of
the study’s hypothesis or research gap.

Data Presentation:

1. Some tables and figures (e.g., heatmaps, haplotype networks) require better
integration into the main text for contextual understanding.

2. Supplementary data references are frequent but insufficiently described, requiring
readers to constantly consult external materials.

Results Interpretation:

1. The low concordance between serological and molecular methods needs a more
detailed discussion. Exploring the potential causes, such as parasite strain
diversity or immune response variability, would add depth to the conclusions.

2. The implications of co-infections with T. dionisii should be further explored,
particularly in the context of cross-reactivity or diagnostic interference.

Language and Style:

1. Some sections contain complex sentence structures and technical jargon that may
hinder accessibility for non-specialist readers. Simplifying language without losing
scientific accuracy would improve readability.

2. Inconsistent use of abbreviations and terms (e.g., PDG, SDG, SAT, kDNA) detracts
from clarity.

Practical Implications and Recommendations:

1. While the manuscript identifies diagnostic shortcomings, it provides limited
actionable recommendations for improving diagnostic protocols in resource-limited
settings.

2. Discussing the feasibility of implementing the “global gold standard” approach in
endemic areas would enhance the translational value of the findings.

To my mind, the manuscript is a significant contribution to Chagas disease
diagnostics. Addressing the identified weaknesses and incorporating the suggested
improvements would enhance its clarity, impact, and utility for both researchers and
public health practitioners. I recommend conducting a thorough language edit to
ensure consistency, reduce redundancy, and improve flow. Finally, I recommend its
acceptance with major revisions.

---

## [Reviewer Report · REVIEWER #2]

This manuscript examines the sensitivity and specificity of current serological and
molecular diagnostic methods for Chagas disease in populations from three regions of
Oaxaca, Mexico. The proposal is particularly relevant in this region due to the
discrepancies in diagnostic results reported in previous studies.

Participants were selected based on clinical and epidemiological criteria (Primary
Diagnosis Group, PDG) and 45 previously diagnosed patients who had not received
treatment (Secondary Diagnosis Group, SDG).

It is unclear whether samples from individuals known to be negative for Chagas
disease were used to monitor cross-contamination during the DNA extraction and
amplification process and to assess the specificity of the PCR reactions.

However, the study’s design posed challenges. It involved administering a large
number of diagnostic tests, both molecular and serological, and not all samples were
evaluated using every method. Consequently, analyzing the data and interpreting the
results effectively took much work.

Which method was considered gold standard?

It is crucial to clearly define the specific type of data used in the statistical
calculations. For example:

1. “Pearson’s correlation coefficient measures the strength and direction of the
linear relationship between two quantitative variables that follow a parametric
distribution.” It is necessary to clarify which values from the ELISA and qPCR tests
were utilized for this analysis. Additionally, the sample size (N) must be
specified, along with a clear definition of the data considered to be the global
standard.

2. Specificity, defined as the ability of a diagnostic test to accurately identify
individuals who do not have the disease, also requires clarification. How was the
specificity of the PCR assays calculated? Were samples from patients with other
diseases, such as leishmaniasis or Chagas disease negative, included in this
assessment?

---

## [Reviewer Report · AUTHORS’ RESPONSE TO THE REVIEWERS]

Responses to the reviewers´ comments on the manuscript ID MIOC-2024-0224 entitled
“Diagnosis of Trypanosoma cruzi infection in Mexican populations: current
conventional serology lacks adequate sensitivity and specificity”. All authors are
grateful for their observations, suggestions and comments to improve the
manuscript.

Reviewer #1

Abstract and Introduction:

1. The abstract lacks clarity in describing the main conclusions and practical
implications of the findings. Refining this section to emphasize the diagnostic
challenges and proposed solutions would enhance its impact.

Response: the section has been edited and findings-challenges and primary solutions
highlighted

2. The introduction is comprehensive but could benefit from a clearer articulation of
the study’s hypothesis or research gap.

Response: The section has been edited and the research gap as well as study
hypothesis more specifically highlighted.

Data Presentation:

1. Some tables and figures (e.g., heatmaps, haplotype networks) require better
integration into the main text for contextual understanding.

Response: The two figures mentioned are fully described and integrated now into the
main text so that contextual understanding is clearer. We have reviewed and expanded
all other tables and figures as well.

2. Supplementary data references are frequent but insufficiently described, requiring
readers to constantly consult external materials.

Response: We have reviewed and edited all references in the main text referring to
data detailed in the supplementary tables so that readers would only need to review
these latter tables if they wish to check additional data details. Due to editorial
limitations on text length, full description of all data would have been
unacceptable, but given the surprising data generated, we felt necessary to make
these available and transparent for our colleagues.

Results Interpretation:

1. The low concordance between serological and molecular methods needs a more
detailed discussion. Exploring the potential causes, such as parasite strain
diversity or immune response variability, would add depth to the conclusions.

Response: We have amplified the discussion regarding concordance, precisely related
to and using multiple details summarized in the supplementary tables (which could
not be adequately highlighted or discussed due to manuscript length limitations).
The failure and limitations of both conventional serology (limited/imprecise immune
response detection) and molecular markers/PCR sensitivity are further mentioned in
this issue.

2. The implications of co-infections with T. dionisii should be further explored,
particularly in the context of cross-reactivity or diagnostic interference.

Response: We have edited and further mentioned the data we have for the few cases of
con-infections, although without having individual infections of T. dionisii in
humans and being able to analyze IR to specific antigens or proteins, it is
impossible with this study to generate further hypotheses. A specific study in these
populations would need to determine if in fact any disease symptoms result from the
dionisii infection, or if is affects T cruzi specific serological responses.

Language and Style:

1. Some sections contain complex sentence structures and technical jargon that may
hinder accessibility for non-specialist readers. Simplifying language without losing
scientific accuracy would improve readability.

Response: We have fully re-edited the text and reduced to the minimum the use of
technical language as long as it does not favor imprecise or generalized reporting
of results, or of data interpretation and discussion.

2. Inconsistent use of abbreviations and terms (e.g., PDG, SDG, SAT, kDNA) detracts
from clarity.

Response: We have attempted to be more precise in the editing and new version

Practical Implications and Recommendations:

1. While the manuscript identifies diagnostic shortcomings, it provides limited
actionable recommendations for improving diagnostic protocols in resource-limited
settings.

Response: We have presented two recommendations for the short and medium term for the
failure of serology: resolve seronegatives using PCR, and develop-test new
multiantigen formulations for serological tests.

Mexico is not a resource-limited country and even first level healthcare for many
infectious pathogens or indeed diseases (ie. Tuberculosis) have not had limited
attention, as is the case of CD. The prevalence is daunting and as it is a chronic
disease, it is too easy to sweep under the PHS attention.

2. Discussing the feasibility of implementing the “global gold standard” approach in
endemic areas would enhance the translational value of the findings.

Response: We have edited the discussion and attempted to present this need,
particularly for clinical settings and with their broadening capacity for molecular
diagnoses. In fact, clinicians are those with increasing requests to test
seronegative patients for T cruzi.

---

## [Reviewer Report · REVIEWER #2]

Which method was considered gold standard?

Response: this is specifically defined in Methods lines 253 to 256, and just prior to
this, specific description of serological, epPCR and qPCR diagnoses.

It is crucial to clearly define the specific type of data used in the statistical
calculations. For example:

1. “Pearson’s correlation coefficient measures the strength and direction of the
linear relationship between two quantitative variables that follow a parametric
distribution.” It is necessary to clarify which values from the ELISA and qPCR tests
were utilized for this analysis.

Response: in Methods, lines 247-252 specifically details that we are analyzing as
positive or negative for each specific test (each serological test, epPCR or qPCR),
for each sample and according to their assigned cohort. lines 257-265 details which
correlations for these same results were calculated. Additionally, the sample size
(N) must be specified, along with a clear definition of the data considered to be
the global standard.

Response: All tables, primary and supplemental, give al sample sizes (N) for each
group.

2. Specificity, defined as the ability of a diagnostic test to accurately identify
individuals who do not have the disease, also requires clarification.

Response: in Methods line 266: True negative/(true negatives + false positives).

How was the specificity of the PCR assays calculated?

Response: specificity for PCR was calculated in the same way as any other, ie. True
negative/(true negatives + false positives).

Were samples from patients with other diseases, such as leishmaniasis or Chagas
disease negative, included in this assessment?

Response: yes, we have a panel of 20 negative samples analyzed for all tests used in
parallel to control for all methods.

---

## [Reviewer Report · SECOND REVIEW ROUND]

REVIEWERS COMMENTS

Reviewer #1

Thank you for the opportunity to review this manuscript. The study addresses a
critical diagnostic challenge in Chagas disease endemic to Mexico and highlights
valuable data on the inadequate performance of conventional serological assays,
while also supporting the utility of molecular methods. The work is relevant to both
clinical and public health contexts. However, some points should be addressed.

Major Limitations

1. The use of a combined “global gold standard” (≥2 serological positives and/or PCR
positives) increases diagnostic sensitivity but risks inflating true positivity,
particularly in settings where molecular positivity may reflect transient
parasitemia or false positives. This definition is operational rather than validated
and may bias performance estimates of individual tests.

2. Despite thorough diagnostic analyses, the study does not correlate test results
with clinical outcomes (e.g., ECG abnormalities, symptom profiles). This makes it
difficult to assess the implications of seronegative-PCR-positive cases in terms of
disease staging or need for treatment.

3. Cohort Characteristics Across Sites are unbalanced. Indeed, differences in age,
sex ratio, and schooling (especially in the SRN site) may confound immune response
patterns and diagnostic performance but are not statistically controlled. This
weakens the strength of comparisons across populations and sites.

4. While the study demonstrates low specificity in serological tests, it does not
sufficiently explore the causes (such as cross-reactivity with T. rangeli, T.
dionisii), or other trypanosomatids. The role of non-specific immune responses or
background reactivity is also not fully addressed.

5. Although Luminex-based profiling of recombinant proteins was performed, the
manuscript does not fully capitalize on this dataset to identify potential antigen
combinations or patterns that might improve future diagnostic test development for
local T. cruzi strains.

Minor Limitations

1. The specificity of PCR assays is reported using 20 negative samples, but it
remains unclear whether samples from individuals with other parasitic diseases
(e.g., leishmaniasis) were included to rigorously test for cross-reactivity.

2. Some abbreviations (e.g., PDG, SDG, SAT, ME) are used inconsistently or without
initial clarification, which may reduce clarity for non-specialist readers.

3. Important findings are often relegated to supplementary tables and figures. This
disperses the narrative and makes it difficult to fully interpret key outcomes
without cross-referencing.

4. The manuscript does not adequately address the feasibility or cost-effectiveness
of implementing PCR-based diagnostics in routine surveillance or primary care
settings in Mexico.

In my oppinion, this study offers compelling evidence for the reevaluation of current
serodiagnostic strategies for Chagas disease in Mexico. The authors are commended
for the scope and rigor of their analyses. Addressing the limitations above would
strengthen the manuscript’s impact and practical relevance.

---

## [Reviewer Report · AUTHORS’ RESPONSE TO THE REVIEWERS]

Responses to the reviewers´ comments on the manuscript ID MIOC-2024-0224.R1 entitled
“Diagnosis of Trypanosoma cruzi infection in Mexican populations: current
conventional serology lacks adequate sensitivity and specificity”. All authors are
grateful for their observations, suggestions and comments to improve the
manuscript.

REVIEWER #1

Thank you for the opportunity to review this manuscript. The study addresses a
critical diagnostic challenge in Chagas disease endemic to Mexico and highlights
valuable data on the inadequate performance of conventional serological assays,
while also supporting the utility of molecular methods. The work is relevant to both
clinical and public health contexts. However, some points should be addressed. Note:
We thank the Reviewer for comments and respond here to each point raised as a major
or minor limitation; we would like to highlight that none of the major points are
considered within the scope of this study´s design and goals to warrant modifying
the text. Concerning minor limitations, we have addressed each point and have made
any additional changes where necessary with change control in the manuscript.

Major Limitations 1. The use of a combined “global gold standard” (≥2 serological
positives and/or PCR positives) increases diagnostic sensitivity but risks inflating
true positivity, particularly in settings where molecular positivity may reflect
transient parasitemia or false positives. This definition is operational rather than
validated and may bias performance estimates of individual tests.

Response: We disagree with the reviewer´s assertion, since we have consistently used
the goal of “T. cruzi infection” which includes “transient parasitemia” (the goal is
not to detect infectiveness) and for which it is well known that false positives may
more probably measure minor cases of false infections due to immunoreactivity of
autoantibodies vs. active infection. If immune response reactivity is not sensitive
as we now know, the only potential method to complement serology, unless there is an
improvement in antigens used for serology, is direct parasite detection, hence our
use of PCR as the most sensitive current method.

2. Despite thorough diagnostic analyses, the study does not correlate test results
with clinical outcomes (e.g., ECG abnormalities, symptom profiles). This makes it
difficult to assess the implications of seronegative-PCR-positive cases in terms of
disease staging or need for treatment.

Response: Correlation with clinical outcomes implies a longitudinal analysis, which
was not the goal for this cohort of the overall study. We are currently writing a
manuscript with a cohort of patients from Yucatan (southeast Mexico) in which the
results show no correlation for seronegativity/PCR positive and cardiac
manifestations compatible with Chagas disease in the participants, but this is a
separate manuscript with different objectives than the one presented here.

3. Cohort Characteristics Across Sites are unbalanced. Indeed, differences in age,
sex ratio, and schooling (especially in the SRN site) may confound immune response
patterns and diagnostic performance but are not statistically controlled. This
weakens the strength of comparisons across populations and sites.

Response: Although age and sex ratio averages are higher and broader for the SRN, an
important consideration to validate preliminary evidence with further analyses of
larger and better matched cohorts/groups included in this study. However, subgroups
within the cohorts (PDG, SDG), were too small to statistically control and therefore
we have taken care not to emphasize or conclude regarding apparent differences in
diagnostic precision, and have emphasized within site differences (PDG vs. SDG) and
overall combined evidence, and not that across populations and sites.

4. While the study demonstrates low specificity in serological tests, it does not
sufficiently explore the causes (such as cross-reactivity with T. rangeli, T.
dionisii), or other trypanosomatids. The role of non-specific immune responses or
background reactivity is also not fully addressed.

Response: With respect, the presence of T. rangeli and T. dionisii in triatomines and
wildlife from these sites has been previously reported (Izeta-Alberdi et al. 2016),
species specific and taxonomic landscape evidence will be published soon; the former
species has not been identified and the latter has, coinciding with co-infections
reported herein. Non-specific immune responses to trypansomatids assumes cross-over
proteomes and potential serological sensitivity bias, not analysed in any study
published related to serological diagnostic performance to date and was not
contemplated as an objective in the present study.

5. Although Luminex-based profiling of recombinant proteins was performed, the
manuscript does not fully capitalize on this dataset to identify potential antigen
combinations or patterns that might improve future diagnostic test development for
local T. cruzi strains.

Response: Evidence that has been cited and published regarding the Luminex RPs
cautions regarding specific molecule reactivity, which is the reason we only analyse
and highlight differences among the proteins related to serological and PCR
reactivity of subgroups based on infection history. We are working collaboratively
to improve diagnostic tests as published in Romer et al, Ricci et al and more
recently Ossowski et al. has already provided evidence of multiple peptide
combinations for improved diagnostic performance.

Minor Limitations 1. The specificity of PCR assays is reported using 20 negative
samples, but it remains unclear whether samples from individuals with other
parasitic diseases (e.g., leishmaniasis) were included to rigorously test for
cross-reactivity.

Response: The primers used to amplify sample DNA have been previously tested on
spiked or patient samples having similar phylogenetic organisms (Leishmania, other
Trypanosomas – few of which have voucher sequences in GenBank) in original and many
subsequent studies (not possible to cite all in references). It was not deemed
necessary to include additional non-T cruzi patient samples in the study design
either for serology or for PCR, and no sequences with greater than 70% identity and
high query cover were identified from any taxonomic proximity to Trypanosoma, none
of them blasted for another species.

2. Some abbreviations (e.g., PDG, SDG, SAT, ME) are used inconsistently or without
initial clarification, which may reduce clarity for non-specialist readers.

Response: All abbreviations have been checked and first usage details are included as
an addendum to these responses. We have modified certain primary citations in the
text to assure all first citations of abbreviations include the complete term. These
changes have been highlighted in yellow in the manuscript.

3. Important findings are often relegated to supplementary tables and figures. This
disperses the narrative and makes it difficult to fully interpret key outcomes
without cross-referencing.

Response: Only key evidence and summarized data are included in primary tables and
figures due to limitations by the journal for the number of tables and figures.
However, we have included disaggregated and details of data not normally published
in other studies for full data availability and transparency.

4. The manuscript does not adequately address the feasibility or cost-effectiveness
of implementing PCR-based diagnostics in routine surveillance or primary care
settings in Mexico.

Response: The goal of the current study was to generate evidence regarding
serological performance for T. cruzi diagnosis in Mexican patients and exposed
populations. Unfortunately, restrictions on length and content of the manuscript
restrict including cost-benefit analyses for different diagnostic protocols vs false
serological negativity, although once we have completed publication of the second
cohort from another region, we will be exploring overall recommendations including
current performance failure and new diagnostic strategies for cost analyses. List of
first citations of Abbreviations: All changes have been highlighted in yellow to
facilitate the reviewer’s correction of the new version of the manuscript. In
Summary epPCR and qPCR are used without giving full name: is it necessary? Chagas
disease in humans (CD) line 65 Mini-exon (ME) line 106 end point polymerase chain
reaction (epPCR) line 108 satellite (SAT) line 109 18S ribosomal DNA (18S rDNA) line
109 discrete typing unit I (DTUI) line 112 (secondary diagnosis group, SDG) line 130
(primary diagnosis group, PDG) line 131-132 quantitative PCR (qPCR) line 135 PHS
(Public Health Service) line 149-150 Sanitary Jurisdiction Heads (JSI, JSII, JSIV
line 151 Santos Reyes Nopala (SRN) line 156 Salina Cruz (SC line 157 Sta Cruz
Papalutla (SCP line 158 serum separation tube-SST Line 177 guanidine – EDTA buffer
(GEB) Line 178 Instituto Nacional de Nutrición y Ciencias Médicas Salvador Zubirán
(INNCMSZ) line 182 enzyme-linked immunosorbent assay (ELISA) line 187 optical
density (OD) line 188 standard deviation (SD) Line 189 green fluorescent protein
(GFP line 197-198 T. cruzi lineage I; TcI line 202 T. cruzi lineage VI; TcVI line
202-203 Instituto de Investigaciones en Ingeniería Genética y Biología Molecular
(INGEBI) line 207-208 kinetoplast DNA (kDNA) line 216 24Sα ribosomal DNA (24S rDNA)
line 219-220 cytochrome b (cyt b) line 225 threshold counts (Ct) line 249
quantitative values (Q), line 255 non-quantitative (NQ) line 256 Triatoma
phyllosoma: T. phyllosoma line 313-314 Triatoma mazzottii ; T. mazzottii line 313
Triatoma dimidiata Hg2: T. dimidiata Hg2 line 314 Triatoma barberi: T. barberi line
315-316 parasites/ml; para/ml) line 319-320 ME lineage I (ME LI) line 338 ME lineage
II (ME LII line 339-340 Trypanosoma dionisii. T. dionisii line 486.

---

## [Reviewer Report · THIRD REVIEW ROUND]

REVIEWERS’ COMMENTS

Reviewer #1

Dear Editor,

I have carefully reviewed the revised version of the manuscript “Diagnosis of
Trypanosoma cruzi infection in Mexican populations: current conventional serology
lacks adequate sensitivity and specificity.” I am satisfied with the authors’
responses to the previous comments and concerns. The revisions have addressed all
major issues, and the manuscript now meets the scientific and editorial standards
required by the journal. In its current form, I recommend acceptance for
publication.

DECISION: ACCEPT SUBMISSION | RECEIVED 10 OCTOBER 2024 | ACCEPTED 19 MAY 2025.